# Constitutively active STING causes neuroinflammation and degeneration of dopaminergic neurons in mice

Eva M Szego[1†], Laura Malz[2†], Nadine Bernhardt[3], Angela Rösen-Wolff[4], Björn H Falkenburger[1,5]*, Hella Luksch[6]

[1]Department of Neurology, TU Dresden, Dresden, Germany; [2]Departments of Neurology & Pediatrics, TU Dresden, Dresden, Germany; [3]Department of Psychiatry, TU Dresden, Dresden, Germany; [4]Department of Pediatrics, TU Dresden, Dresden, Germany; [5]Deutsches Zentrum für Neurodegenerative Erkrankungen, Dresden, Germany; [6]Department of Pediatrics, TU Dresden, Dresden, Germany

**\*For correspondence:**
Bjoern.Falkenburger@dzne.de

[†]These authors contributed equally to this work

**Competing interest:** The authors declare that no competing interests exist.

**Abstract** Stimulator of interferon genes (STING) is activated after detection of cytoplasmic dsDNA by cGAS (cyclic GMP-AMP synthase) as part of the innate immunity defence against viral pathogens. STING binds TANK-binding kinase 1 (TBK1). TBK1 mutations are associated with familial amyotrophic lateral sclerosis, and the STING pathway has been implicated in the pathogenesis of further neurodegenerative diseases. To test whether STING activation is sufficient to induce neurodegeneration, we analysed a mouse model that expresses the constitutively active STING variant N153S. In this model, we focused on dopaminergic neurons, which are particularly sensitive to stress and represent a circumscribed population that can be precisely quantified. In adult mice expressing N153S STING, the number of dopaminergic neurons was smaller than in controls, as was the density of dopaminergic axon terminals and the concentration of dopamine in the striatum. We also observed alpha-synuclein pathology and a lower density of synaptic puncta. Neuroinflammation was quantified by staining astroglia and microglia, by measuring mRNAs, proteins and nuclear trans-location of transcription factors. These neuroinflammatory markers were already elevated in juvenile mice although at this age the number of dopaminergic neurons was still unaffected, thus preceding the degeneration of dopaminergic neurons. More neuroinflammatory markers were blunted in mice deficient for inflammasomes than in mice deficient for signalling by type I interferons. Neurodegeneration, however, was blunted in both mice. Collectively, these findings demonstrate that chronic activation of the STING pathway is sufficient to cause degeneration of dopaminergic neurons. Targeting the STING pathway could therefore be beneficial in Parkinson's disease and further neurodegenerative diseases.

## Editor's evaluation

STING has recently been implicated in the pathogenesis of neurodegenerative diseases. In the current work, the authors analysed a mouse model that expresses the constitutively active STING variant N153S and demonstrated that activation of the STING pathway via downstream signalling by type I interferons (Ifnar1-/-) or inflammasome activation (Casp1-/-) is sufficient to induce degeneration of dopaminergic neurons. Moreover, STING activation is sufficient to induce α-synuclein pathology in Parkinson's disease. This finding is of interest to the broad readership of *eLife*.

**eLife digest** Neurodegenerative conditions such as Alzheimer's and Parkinson's diseases are characterised by neurons getting damaged and dying. Many factors contribute to this process, but few can be effectively controlled by therapies. Interestingly, previous studies have highlighted that inflammation, a process normally triggered by foreign agents or biological damage, is often associated with neurons degenerating. However, it is unclear whether these responses are the cause or the consequence of brain cell damage.

In injured neurons, the genetic information normally contained inside a dedicated cellular compartment can start to leak into the surrounding parts of the cell. This damage triggers an inflammatory response through the STING pathway, a mechanism previously implicated in the onset of Parkinson's disease. In these patients, the neurons that produce the signalling molecule dopamine start to die, leading to difficulty with movement. Whether STING can directly cause this neuronal loss remains unknown.

To answer this question, Szegö, Malz et al. genetically engineered mice in which the STING pathway is permanently activated. The animals had fewer dopamine-producing neurons and accumulated harmful clumps of proteins; both these biological features are characteristic signs of Parkinson's disease. Crucially, signs of inflammation were present before neurons started to show damage, suggesting that inflammatory responses could cause neurodegeneration. Further experiments revealed that STING triggers several molecular cascades; blocking one only of these pathways did not keep the neurons healthy.

Neurodegenerative diseases are a growing concern around the world. The results from Szegö, Malz et al. suggest that preventing prolonged inflammatory may reduce the risk of neurodegeneration. If further research confirms these findings, in particular in humans, well-known treatments against inflammation could potentially become relevant to fight these conditions.

## Introduction

Inflammation is recognized increasingly as a major driver for neurodegenerative diseases, including Parkinson's disease (PD) (*Harms et al., 2021*; *Hirsch and Standaert, 2021*). The stimulator of interferon genes (STING) responds to cytoplasmic dsDNA as part of the innate immunity defence against viral pathogens (*Paul et al., 2021*). STING is activated by cyclic GMP-AMP (cGAMP) produced by cGAMP synthase (cGAS) upon binding to dsDNA (*Motwani et al., 2019*). This pathway can be activated by pathogen-derived nucleic acids but also by cytoplasmic self-DNA (*Chen et al., 2016*; *Li and Chen, 2018*; *Motwani et al., 2019*). STING binds TANK-binding kinase 1 (TBK1) and activates the transcription factors interferon regulatory factor 3 (IRF3) (*Chen et al., 2021*) and nuclear factor 'kappa-light-chain-enhancer' of activated B-cells (NF-κB) (*Liu et al., 2014*). The resulting increased production of type I interferons (IFN) and pro-inflammatory cytokines induce the activation of inflammasomes and inflammation (*Hopfner and Hornung, 2020*). STING, however, can activate inflammasomes directly (*Wang et al., 2020*). Inflammasomes are major signalling hubs that activate caspase-1 and control the bioactivity of pro-inflammatory cytokines of the interleukin (IL)–1 family (*Gaidt et al., 2017*; *Schroder and Tschopp, 2010*).

The cGAS-STING pathway has been implicated in PD pathogenesis. Mice deficient for the PD-associated proteins PTEN-induced kinase 1 (PINK) and parkin show a STING-mediated inflammatory phenotype after exhaustive exercise (*Sliter et al., 2018*). This phenotype results from the PINK/parkin-dependent failure to degrade damaged mitochondria and the consecutive accumulation of mitochondrial DNA in the cytosol. Indeed, inflammatory markers like IL-6 were also observed in human patients with parkin mutations (*Sliter et al., 2018*). Similarly, cytoplasmic accumulation of TDP-43, which constitutes the most common pathological finding in sporadic amyotrophic lateral sclerosis (ALS), triggers the release of mitochondrial DNA and the consecutive, STING-dependent neuroinflammation and neurodegeneration (*Yu et al., 2020*). In addition, mutations in TBK1 are associated with familial ALS (*Freischmidt et al., 2015*). Finally, activation of inflammasomes formed by the NOD-, LRR- and pyrin domain-containing protein 3 (NLRP3) has been linked to the progression of several neurodegenerative diseases (*Heneka et al., 2018*).

Neuroinflammation receives growing attention in neurodegenerative diseases because it represents a promising therapeutic target. Designing such therapies, however, requires to determine the specific effects of individual components of this highly interconnected signalling network, which responds to diverse stimuli and involves many different cell types. Previous studies have demonstrated that depleting or blocking STING can alleviate neurodegeneration in models of PD (*Sliter et al., 2018*) and ALS (*Yu et al., 2020*). To complement these observations and provide a model to study the signalling downstream of STING, we wanted to determine whether specific activation of STING is sufficient to cause neurodegeneration. In order to test this, we used a mouse model with heterozygous expression of the STING genetic variant N153S (*Luksch et al., 2019*). Constitutively active STING mutants cause an autoinflammatory disease in humans termed STING-associated vasculopathy with onset in infancy (SAVI) (*Crow and Casanova, 2014*; *Liu et al., 2014*). SAVI is characterized by systemic inflammation with acral vasculitis, T cell lymphopenia, and interstitial pulmonary disease. Major features of systemic inflammation in SAVI are recapitulated in STING N153S knock-in mice (*Luksch et al., 2019*; *Siedel et al., 2020*). For simplicity, we refer to these as STING ki mice here and use the term STING WT for the corresponding wild type littermate controls. In these mice, we determined the extent of neuroinflammation and quantified the integrity of dopaminergic neurons that project from the substantia nigra pars compacta to the striatum. We focused on dopaminergic neurons because they have been the centre of our investigations (*Falkenburger et al., 2001*; *Komnig et al., 2018*; *Szegö et al., 2022*; *Szego et al., 2012*), and because STING has been implicated in the pathogenesis of PD (*Hinkle et al., 2022*; *Sliter et al., 2018*). Furthermore, we measured alpha-synuclein (aSyn) pathology, which constitutes the second neuropathological feature of PD next to the degeneration of dopaminergic neurons. To determine which of the known downstream signalling pathways contribute to STING-induced neurodegeneration, we used mice deficient for type I IFN receptor (*Ifnar1*) or caspase 1 (*Casp1*).

## Results

### Neuroinflammation and degeneration of dopaminergic neurons in mice with constitutive STING activation

To characterize the neuronal phenotype in adult STING ki mice, we assessed neuroinflammation and the integrity of the dopaminergic nigrostriatal system. First, we stained striatal sections for Iba1 to determine the activation of microglia. The Iba1 positive area fraction was 9-fold higher in STING ki mice than in STING WT (*Figure 1A and C*), accompanied by an 3.8-fold increase in the number of Iba1-positive cells (*Figure 1—figure supplement 1A*). Activation of astroglia, as determined by GFAP staining, was 26-fold higher in STING ki mice than in STING WT (*Figure 1B and D*), and the number of astroglial cells increased to 6.7-fold (*Figure 1—figure supplement 1B*). Similarly, in the substantia nigra (SN), activation of astroglia (*Figure 1—figure supplement 1C, F, H*) and microglia (*Figure 1—figure supplement 1C, G, I*) increased as well. Thus, STING ki mice exhibit a strong neuroinflammatory phenotype, consistent with increased systemic inflammation (*Luksch et al., 2019*).

Next, we asked whether the chronic neuroinflammation in the STING ki mice is associated with the degeneration of dopaminergic neurons. Somata of dopaminergic neurons in the SN (*Figure 1E*) and dopaminergic axon terminals in the striatum (*Figure 1F*) were identified by staining for tyrosine hydroxylase (TH). The SN of STING ki mice contained significantly fewer TH-positive neurons than the SN of STING WT (*Figure 1G* and *Figure 1—figure supplement 1D*). Similarly, the density of dopaminergic axon terminals (fibers) in the striatum was lower in STING ki mice than in STING WT (*Figure 1H*). Accordingly, the concentration of dopamine in the striatum was lower in STING ki mice than in STING WT mice (*Figure 1I*). The concentration of dopamine metabolites was higher in STING ki mice (*Figure 1—figure supplement 1E*), suggesting increased dopamine turnover in STING ki mice - as commonly observed with degeneration of dopaminergic axon terminals. Taken together, these findings demonstrate that the integrity of nigrostriatal dopaminergic neurons is compromised in STING ki mice.

### Neuroinflammation without degeneration of the dopaminergic neurons in juvenile mice with constitutive STING activation

In order to explore whether the compromised integrity of dopaminergic neurons is a consequence of a prolonged neuroinflammation, we next analysed brain sections of juvenile (5-week-old) STING

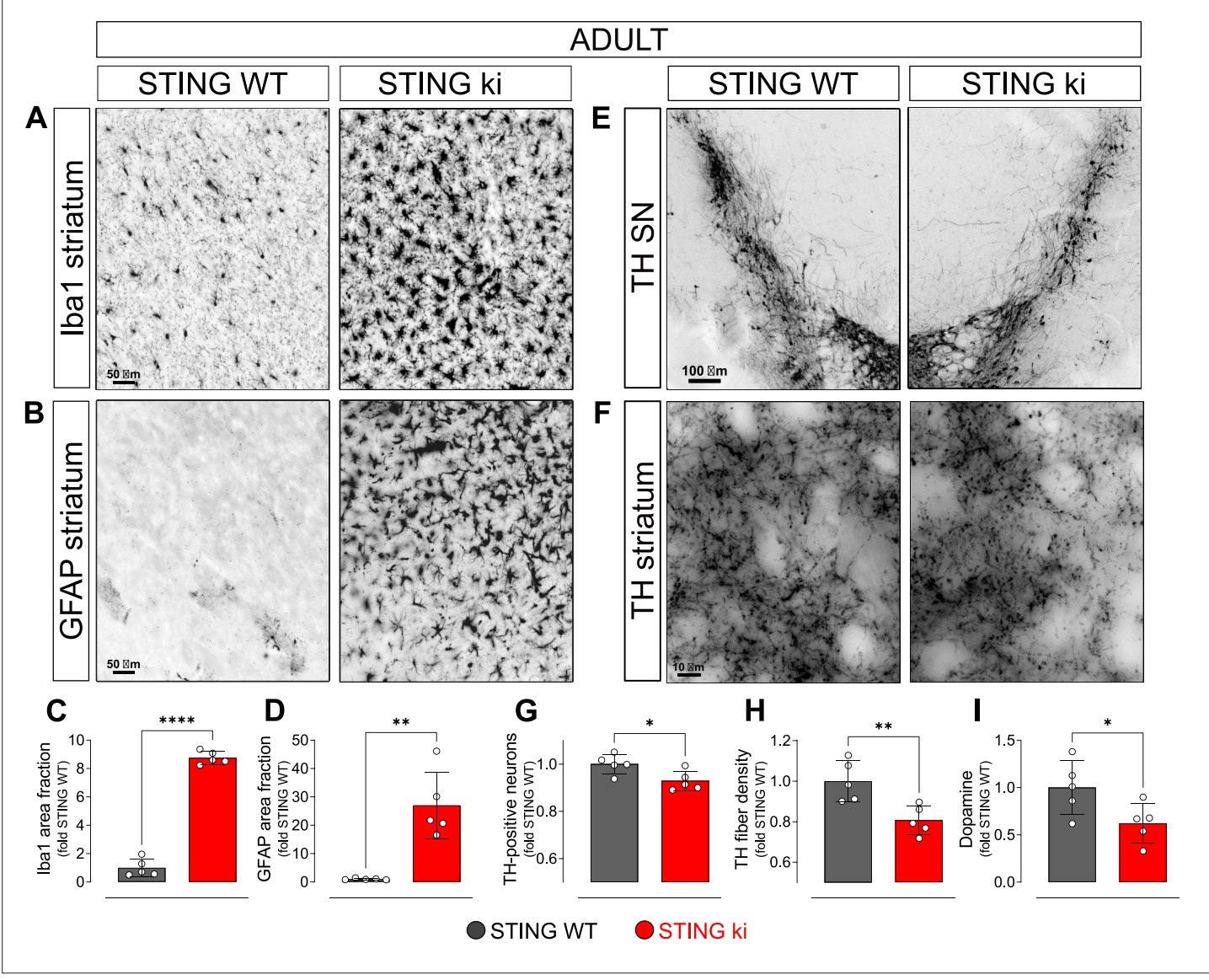

**Figure 1.** Constitutive STING activation induces neuroinflammation and neurodegeneration in adult mice. (**A**) Representative images of striatal sections from STING WT and STING ki mice stained for the microglia marker Iba1. Scale bar: 50 µm. (**B**) Representative images of striatal sections stained for the astroglia marker GFAP. Scale bar: 50 µm. (**C**) Representative images of midbrain sections containing the substantia nigra (SN) from STING WT and STING ki mice (stitched from two microscopy fields) stained for tyrosine hydroxylase (TH). Scale bar: 100 µm. (**D**) Representative images of striatal sections stained for TH from STING WT and STING ki mice. Scale bar: 10 µm. (**E**) Area fraction positive for Iba1, normalized to the mean of STING WT mice. Markers represent individual animals (black: STING WT animals, red: STING ki animals). Lines represent mean ± SD. Comparison by t-test (***: p=0.0007, n=5). Graph showing the counted numbers of Iba1-positive neurons is on *Figure 1—figure supplement 1A* (**F**) Area fraction positive for GFAP, normalized to the mean of STING WT mice (**: p=0.0011; t-test, n=5). Graph showing the counted numbers of GFAP-positive neurons is on *Figure 1—figure supplement 1B*. (**G**) Number of TH-positive neurons (*: p=0.0257; t-test, n=5). Graph showing the counted numbers of dopaminergic neurons is on *Figure 1—figure supplement 1D*. (**H**) Area fraction positive for TH (**: p=0.0081; t-test, n=5). (**I**) Concentration of dopamine (*: p=0.0448; t-test, n=5) in striatal lysates from STING WT and STING ki animals, normalized to the mean concentration in STING WT. Graph showing quantification of the dopamine metabolites is in *Figure 1—figure supplement 1E*.

The online version of this article includes the following figure supplement(s) for figure 1:

**Figure supplement 1.** Neuroinflammation and neurodegeneration in adult mice.

ki and STING WT mice (*Figure 2*). Microglia was already significantly activated in juvenile STING ki mice. The area fraction of the Iba1 staining was twofold higher in STING ki mice than in STING WT (*Figure 2A and E*). Similarly, the area fraction of GFAP signal was 14-fold higher in STING ki mice than in STING WT (*Figure 2B and F*), suggesting activation of astroglia in juvenile STING ki mice. However,

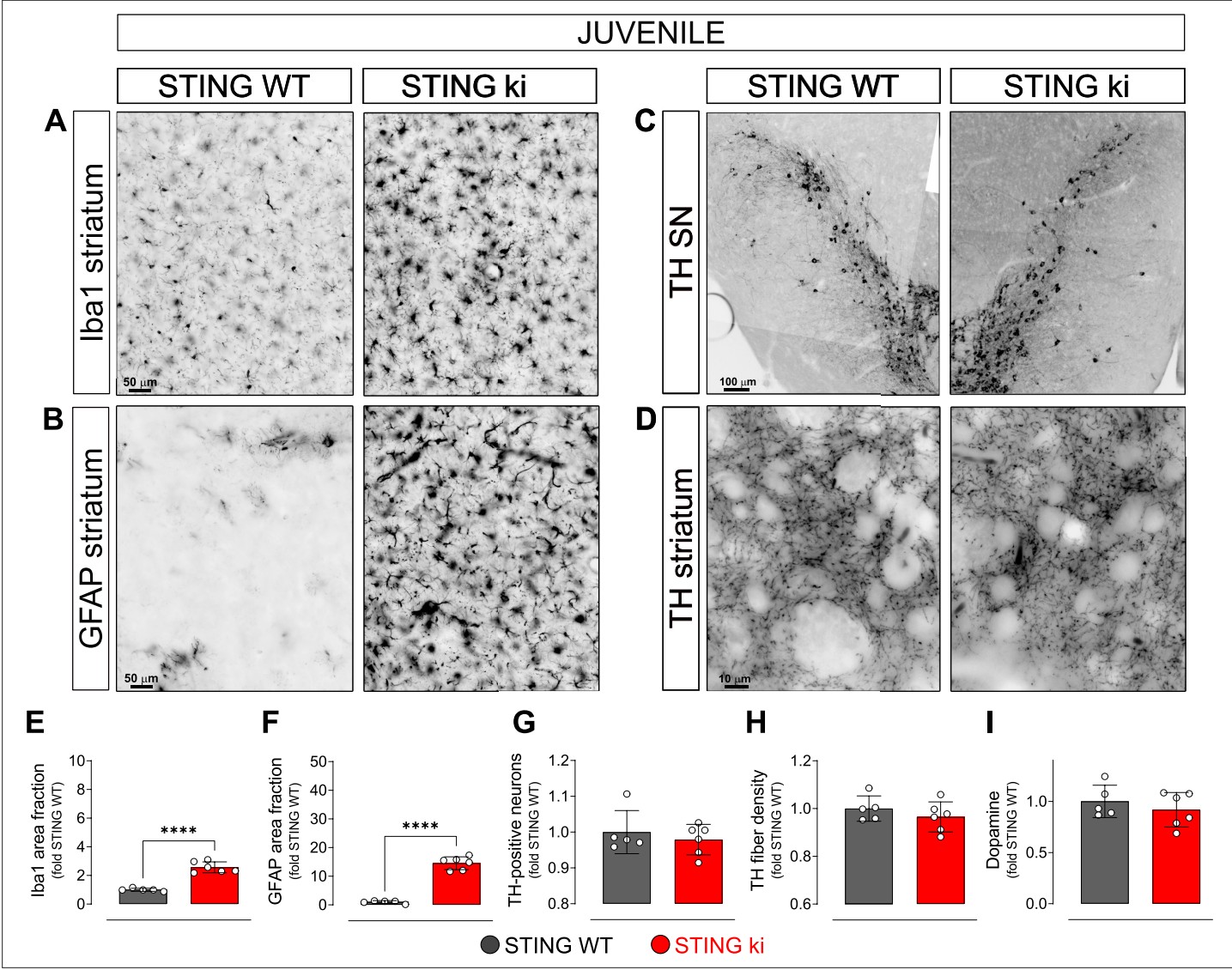

**Figure 2.** Neuroinflammation without neurodegeneration in juvenile mice with constitutive STING activation. (**A**) Representative images of striatal sections stained for the microglia marker Iba1 from 5-week-old STING WT and STING ki mice. Scale bar: 50 µm (**B**) Representative images of striatal sections stained for the astroglia marker GFAP from 5-week-old STING WT and STING ki mice. Scale bar: 50 µm (**C**) Representative images of midbrain sections containing the substantia nigra (SN, stitched from two microscopy fields) stained for tyrosine hydroxylase (TH) from 5-week-old STING WT and STING ki mice. Scale bar: 100 µm (**D**) Representative images of striatal sections stained for TH from 5-week-old STING WT and STING ki mice. Scale bar: 10 µm. (**E**) Area fraction positive for Iba1, normalized to the mean of STING WT (***: p=0,0009; t-test, n=5–6). (**F**) Area fraction positive for GFAP, normalized to the mean of STING WT brains (***: p=0.0007; t-test, n=5–6). (**G**) Number of TH-positive neurons (mean ± SD; t-test). Graph showing the counted numbers of dopaminergic neurons is on *Figure 2—figure supplement 1A*. (**H**) Area fraction positive for TH (mean ± SD, t-test, n=5–6). (**I**) Dopamine concentration in striatal lysates from 5-week-old STING WT and STING ki mice, measured by HPLC and normalized to the mean of STING WT (mean ± SD, t-test, n=5–6). Dopamine metabolites are in *Figure 2—figure supplement 1B*.

The online version of this article includes the following figure supplement(s) for figure 2:

**Figure supplement 1.** Neuroinflammation without neurodegeneration in juvenile mice.

the number of TH-positive neurons in the substantia nigra was not different between juvenile STING ki mice and STING WT (*Figure 2C and G* and *Figure 2—figure supplement 1A*). Similarly, striatal axon terminals (*Figure 2D and H*) and the concentrations of striatal dopamine and its metabolites (*Figure 2I*, and *Figure 2—figure supplement 1B*) were not different between STING ki mice and STING WT.

Taken together, these findings demonstrate that the compromised integrity of the nigrostriatal system in adult STING ki mice (*Figure 1*) represents an adult-onset neurodegeneration and not a developmental defect. Given that activation of microglia and astroglia in STING ki mice precedes degeneration of dopaminergic neurons, STING-induced neuroinflammation could contribute to the neurodegeneration.

## aSyn pathology and synaptic defects in the striatum of STING ki mice

In PD, degeneration of dopaminergic neurons is associated with aSyn pathology. We therefore analysed aSyn pathology in the striatum and in the SN of STING ki mice. First, we measured the amount of aSyn protein phosphorylated at serine 129 (paSyn), which is considered one of the major pathological forms of aSyn (*Anderson et al., 2006*; *Fujiwara et al., 2002*; *Samuel et al., 2016*). Both striatal and nigral lysates of adult STING ki mice contained a substantial amount of paSyn (*Figure 3A and B*), which was barely detectable in STING WT. The amount of total aSyn was lower in STING ki mice in both brain regions (*Figure 3—figure supplement 1*) and the ratio of paSyn to total aSyn increased in STING ki mice (*Figure 3B*) – as commonly observed in synucleinopathy models and PD patients (*Anderson et al., 2006*; *Chatterjee et al., 2020*; *Fujiwara et al., 2002*; *Szegö et al., 2022*). To further characterize aSyn pathology in STING ki mice, we assayed the Triton X-100 solubility of αSyn in lysates prepared from the striatum or from the SN (*Figure 3C*). In lysates prepared from STING ki animals, αSyn was detected both in the Triton X-100 insoluble fraction (*Figure 3C* upper panel) and in the Triton X-100 soluble fraction (*Figure 3C* lower panel). In lysates prepared from STING WT animals, αSyn was barely detectable in the Triton X-100 insoluble fraction. Consequently, the ratio of insoluble to soluble aSyn was strongly increased in STING ki mice (*Figure 3D*).

As an alternative approach to detect aggregated proteins, we used Thioflavin S (ThioS), which binds to the characteristic β-sheet conformation of amyloid-containing proteins, including aSyn (*Froula et al., 2019*; *Neumann et al., 2002*). The number of cells with ThioS-positive inclusions was higher in the striatum of adult STING ki mice than in STING WT (*Figure 3E and F*), consistent with the findings from the aSyn immunoblots (*Figure 3A and B*).

Since dopamine depletion and aSyn pathology can compromise synaptic integrity, we next quantified the density of synapses in the striatum. Presynaptic puncta were detected by staining against synapsin; post-synaptic puncta were detected by staining against homer (*Figure 3G–I*). The density of synapsin puncta was 18% lower in adult STING ki than in STING WT (*Figure 3H*), the density of post-synaptic puncta was 9% lower in STING ki than in STING WT mice (*Figure 3I*). In summary, we observed aSyn pathology in the striatum and in the SN, and a reduced density of synapses in the striatum of adult STING ki mice.

## Type I IFN signalling and NF-κB and inflammasome-dependent signalling are activated in the brain of STING ki mice

In order to analyse the signalling pathways by which constitutively active STING causes degradation of dopaminergic neurons and aSyn pathology, we examined the expression of selected interferon-stimulated genes (ISGs) by quantitative real time PCR in the SN and in the striatum of adult STING ki and STING WT mice (*Figure 4*). Expression of the interferon-induced protein 44 (*Ifi44*, *Figure 4A and G*) and the interferon-induced GTP-binding protein *Mx1* (*Figure 4B and H*) was higher in the striatum of STING ki mice (*Figure 4G and H*); this difference was not statistically significant in the SN (*Figure 4A and B*). Similarly, expression of *Sting1* was increased in the striatum of STING ki mice, but not in the SN (*Figure 4C, I*).

cGAS/STING activation also leads to the activation of the NF-κB and inflammasome pathways (*Balka et al., 2020*; *Balka and De Nardo, 2021*; *Wang et al., 2020*). We therefore analysed induction of the NF-κB and inflammasome pathways by quantitative real time PCR of the downstream mediators, tumour necrosis factor alpha (*Tnfa*), interleukin 1 beta (*Il1b*), and caspase-1 (*Casp1*). The average expression of *Tnfa* was higher in STING ki mice compared to STING WT mice, but the difference was not statistically significant (*Figure 4D and J*). In contrast, expression *Il1b* and *Casp1* was significantly higher both in SN and in striatum (*Figure 4E, F, K and L*). Upregulation of inflammatory mediators was also observed in juvenile mice and in cortical lysates (*Figure 4—figure supplement 1*). Collectively, these findings are consistent with differential activation of IFN-dependent and IFN independent signalling between brain regions, but further work will be needed to confirm this.

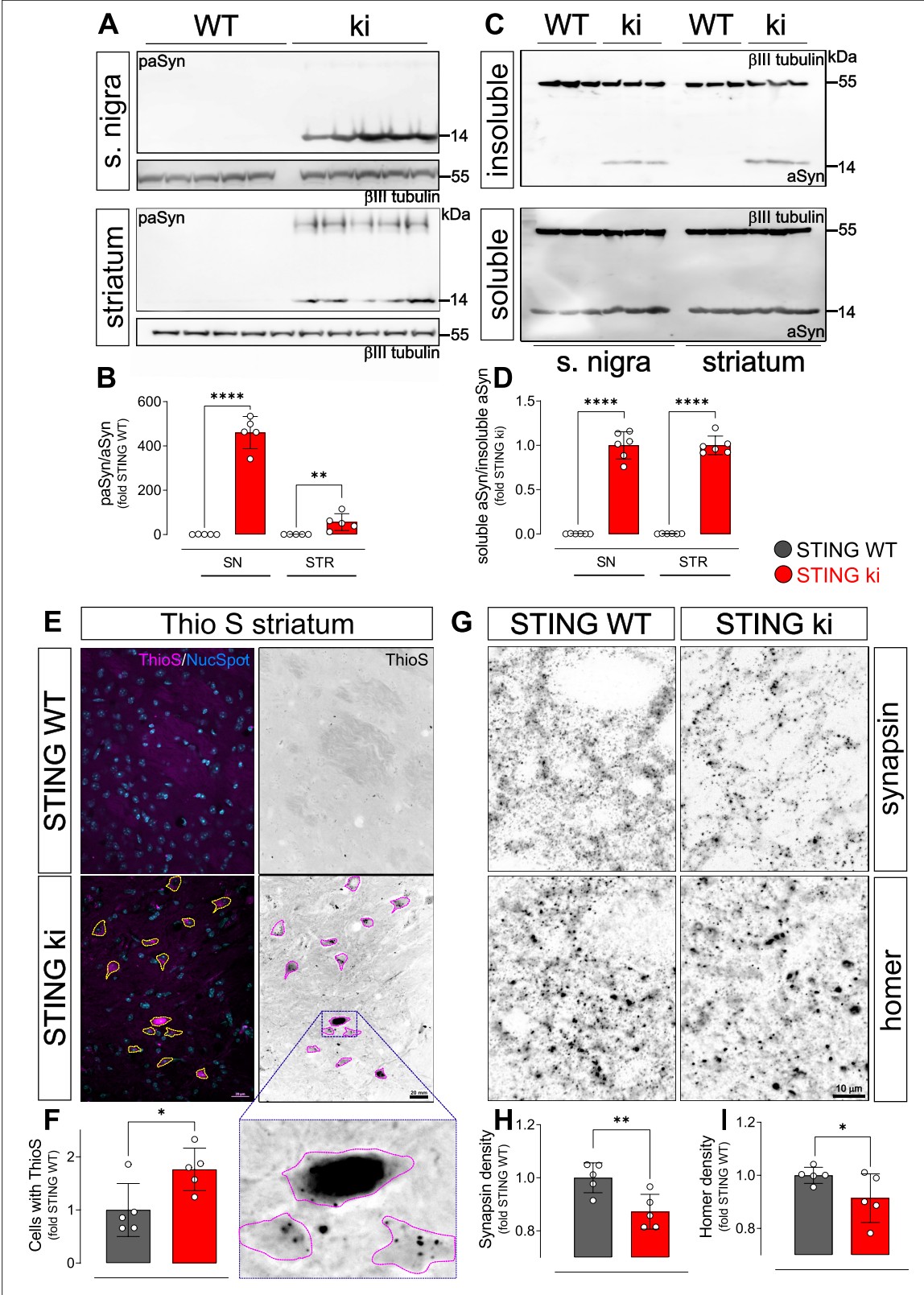

**Figure 3.** Constitutive STING activation induces alpha-synuclein pathology and synapse loss in adult mice. (**A**) Representative western blot images showing phosphorylated alpha-synuclein (S129; paSyn) and the loading control βIII-tubulin detected from the substantia nigra (upper panel) and striatum (lower panel). Total levels of aSyn detected from the same membranes are shown on *Figure 3—figure supplement 1A, B*. (**B**) Ratio of paSyn and total aSyn signals, expressed as relative to the mean of STING WT (substantia nigra (SN): p=0.000065; striatum: p=0.019; t-test, n=5).

*Figure 3 continued on next page*

*Figure 3 continued*

(**C**) Representative western blot images showing aSyn and βIII-tubulin detected from the Triton X-100 insoluble (upper panel) and soluble (lower panel) fractions prepared from the substantia nigra or from the striatum. (**D**) Ratio of aSyn signals detected in the Triton X-100 soluble and insoluble fractions, expressed as relative to the mean of STING ki (substantia nigra (SN): p=0.00001; striatum: p<0.00001; t-test, n=6). (**E**) Representative images of striatal sections from 20-week-old STING WT and STING ki mice stained with Thioflavin S (magenta) and nuclear sdye (blue) on the composite images, and ThioS BW. Scale bar: 20 μm. (**F**) Number of cells with inclusions positive for Thioflavin S (ThioS) per mm$^2$ (*: p=0.0141; t-test, n=5). (**G**) Representative images of striatal sections from 20-week-old STING WT and STING ki mice stained for the presynaptic marker synapsin (upper panel) or for the post-synaptic marker homer (lower panel). Scale bar: 10 μm. (**H–I**) Area fraction positive for synapsin (H, p=0.0053) or homer (I, p=0.0408) (mean ± SD; t-test, n=5).

The online version of this article includes the following source data and figure supplement(s) for figure 3:

**Figure supplement 1.** Uncropped membranes showing α-Synuclein signal detected from whole cell lysates of striatum and substantia nigra.

**Figure supplement 1—source data 1.** Uncropped membranes showing α-Synuclein signal detected from whole cell lysates of striatum and substantia nigra.

## Pro-inflammatory transcription factors translocate to the nucleus in STING ki mice

Nuclear trafficking is critical for the function of transcription factors such as STAT3 and NF-κB (*Balka and De Nardo, 2021*; *Noguchi et al., 2013*). To confirm the activation of the type I IFN and NF-κB-dependent pathways in STING ki mice by another method, we therefore quantified the number of nuclei positive for phosphorylated STAT3 (pSTAT3, *Figure 5A and C*) and for the NF-κB p65 subunit (*Figure 5B and D*) in the striatum of juvenile and adult mice. There were only very few pSTAT3-positive nuclei in juvenile and adult STING WT mice (*Figure 5A*). In juvenile STING ki mice, the average number of pSTAT3-positive nuclei was fourfold higher than in juvenile STING WT mice (*Figure 5C*, p=0.05678, twoway ANOVA). In adult STING ki mice, the number of pSTAT3-positive nuclei was 15-fold higher than in adult STING WT (p=0.00004). The number of NF-κB-positive nuclei was threefold higher in juvenile STING ki mice than in juvenile STING WT (*Figure 5B* and, p=0.009). In adult STING ki mice, the number of NF-kB-positive nuclei was sixfold higher than in adult STING WT (*Figure 5D*, p=0.007). Since many of the pSTAT3-positive nuclei were negative for both Iba1 and GFAP (*Figure 5A*), we next stained striatal sections for the neuronal marker NeuN to determine whether the pSTAT3-positive nuclei are neurons (*Figure 5—figure supplement 1A, B*). Indeed, 67% of NeuN-positive neurons were also positive for pSTAT3 in STING ki animals and only 1% in STING WT (*Figure 5—figure supplement 1B*). In fact, neuronal nuclei represent the majority of nuclear pSTAT3 signal (85% in STING WT and 95% in STING ki, *Figure 5—figure supplement 1C*).

Taken together, our results show nuclear translocation of pSTAT3 and NF-κB in the striatum of STING ki mice, consistent with a robust activation of type I IFN and NF-κB signalling in STING ki mice.

We next assessed nuclear translocation of interferon regulatory factor 3 (IRF3) in striatal lysates obtained from adult STING WT and STING ki mice using subcellular fractionation (*Figure 6A* and *Figure 6—figure supplement 1*). Proper separation of cytosol and nuclei was confirmed by observing HDAC6 exclusively in the nuclear fraction and GAPDH exclusively in the cytosolic fraction. IRF3 was mainly found in the cytosolic fraction of STING WT lysates, but mainly in the nuclear fraction of STING ki lysates (quantified in *Figure 6B* and *Figure 6—figure supplement 1D*). This finding suggests that IRF3 translocates to the nucleus in STING ki mice, similar to pSTAT3 and NF-κB (*Figure 5*). Furthermore, we investigated the contribution of the type I IFN and inflammasome signaling pathways to neuroinflammation in STING ki mice. In order to test the involvement of the IFN dependent pathway, we crossed STING ki mice with mice deficient for type I IFN receptor (*Ifnar1*[-/-]); in order to test the involvement of the inflammasome pathway, we crossed STING ki mice with mice deficient for caspase-1 (*Casp1*[-/-]). Nuclear translocation of IRF3 in STING ki mice was preserved in animals deficient for type I IFN receptor (*Ifnar1*[-/-]), consistent with the notion that IFN signal downstream of IRF3. IRF3 translocation was also preserved in mice deficient for caspase-1 (*Casp1*[-/-]).

## Inflammasomes are activated in STING ki mice

In order to determine whether inflammasomes are activated in STING ki mice, we first measured the abundance of NLRP3 protein in striatal lysates (*Figure 6C*, quantified in D, and *Figure 6—figure supplement 1F*). NLRP3 immunoreactivity was higher in lysates of STING ki mice than in STING WT.

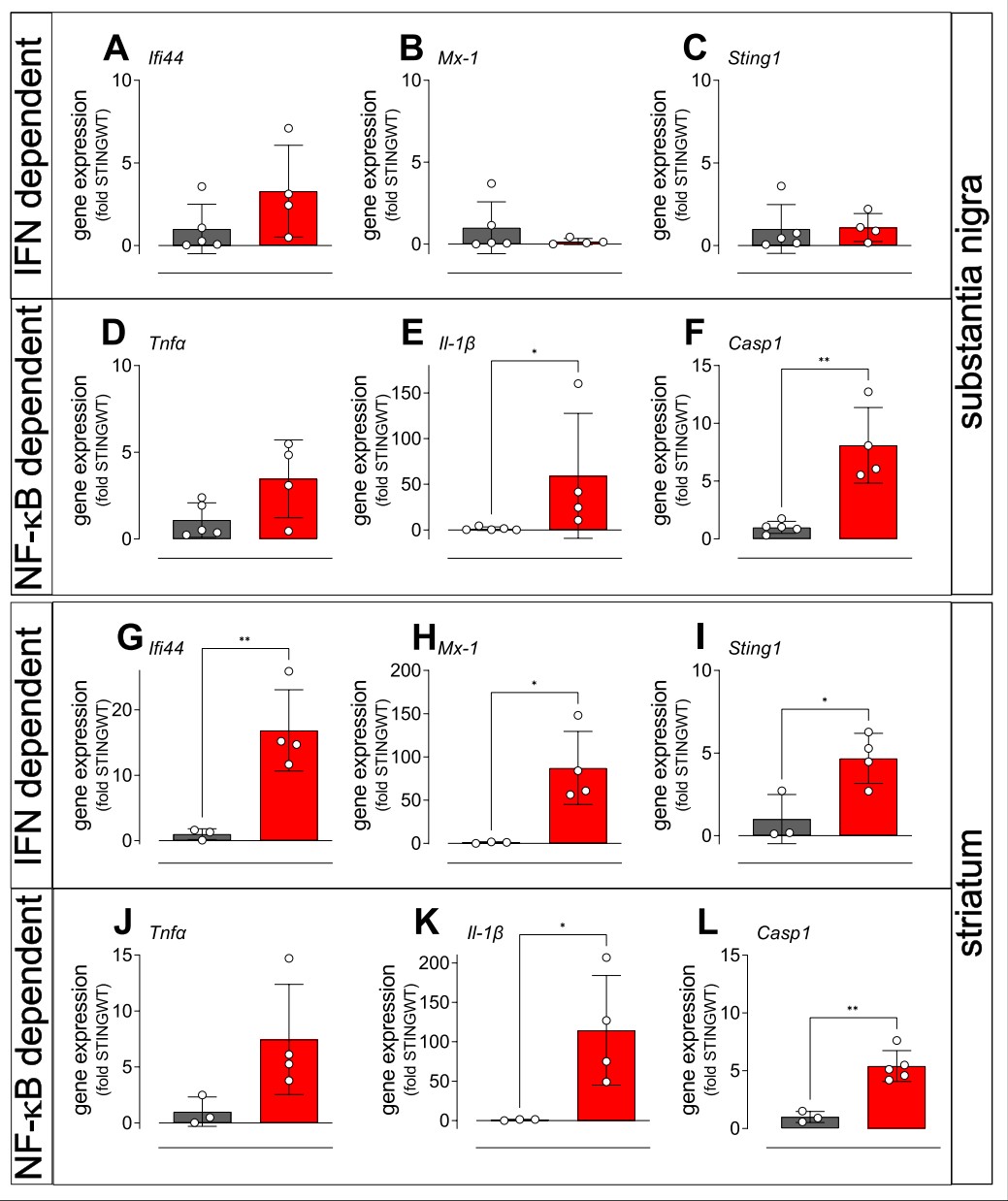

**Figure 4.** Activation of IFN and NF-$\kappa$B/inflammasome related genes in the striatum and SN of STING ki mice. (**A–C**) Expression of ISGs in the substantia nigra of STING WT and STING ki mice. (**A**) *Ifi44* (p=0.1552, n=4–5), (**B**) *Mx1* (Mann Whitney test, p=0.9048, n=4–5), (**C**) *Sting1* (P=0.9184, n=4–5). (**D–F**) Expression of NF-$\kappa$B/inflammasome related genes in the substantia nigra of STING WT and STING ki mice. (**D**) *Tnfa* (p=0.0691, n=4–5), (**E**) *Il1b* (Mann Whitney test, *: p=0.0159, n=4–5), (**F**) *Casp1* (**: p=0.0018, n=4–5). (**G–L**) Expression of ISGs in the striatum of STING WT and STING ki mice. (**G**) *Ifi44* (**: p=0.0078, n=3–4), (**H**) *Mx1* (*: p=0.0183, n=3–4), (**I**) *Sting1* (*: p=0.024, n=3–4). (**J–L**) Expression of NF-$\kappa$B/inflammasome related genes in the striatum of STING WT and STING ki mice. (**J**) *Tnfa* (n=3–4), (**K**) *Il1b* (*: p=0.0397, n=3–4), (**L**) *Casp1* (**: p=0.0017, n=3–5). Markers represent individual animals, bars represent mean ± SD. Analysis was t-test, if not indicated otherwise.

The online version of this article includes the following figure supplement(s) for figure 4:

**Figure supplement 1.** Expression of IFN and NF-$\kappa$B and inflammasome dependent genes in the cortex of juvenile and adult STING ki mice.

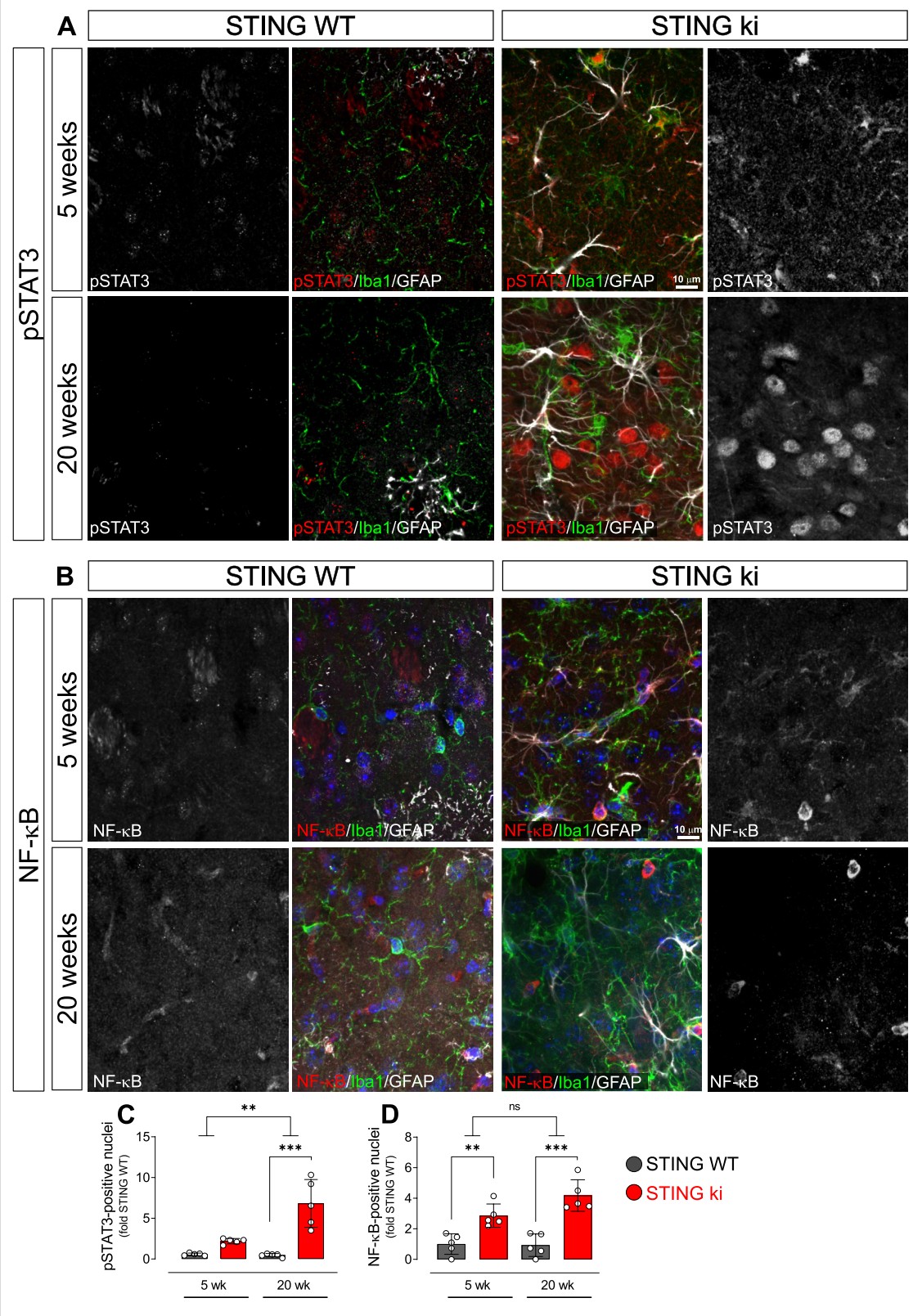

**Figure 5.** Nuclear translocation of pSTAT3 and NF-κB in the striatum of 5-week-old and 20 week-old STING WT and STING ki mice. (**A**) Representative images of striatal sections from 5-week-old (upper images) and 20-week-old (lower images) STING WT and STING ki mice stained for Iba1 (green), GFAP (white) and phosphorylated-STAT3 (pSTAT3; red). Images show color coded merged channels (center) and in addition pSTAT3 staining in grayscale (left and right). Scale bar: 10 μm. (**B**) Representative images of striatal sections from 5-week-old (upper images) and 20 week-old (lower images)

*Figure 5 continued on next page*

*Figure 5 continued*

STING WT and STING ki mice stained for Iba1 (green), GFAP (white), and NF-$\kappa$B (red). NF-kB staining is shown in grey in separate images. Scale bar: 10 μm. (**C**) Number of pSTAT3-positive nuclei/mm³ (***: p=0.00004; **: p=0.0025 for the interaction; two-way ANOVA, Bonferroni post-hoc test, n=5). (**D**) Number of NF-kB-positive nuclei/mm³ (**: p=0.009; ***: p=0.0007; mean ± SD; two-way ANOVA, Bonferroni post-hoc test, n=5).

The online version of this article includes the following figure supplement(s) for figure 5:

**Figure supplement 1.** Uncropped membranes showing the levels of inflammatory proteins in the striatum.

This difference was preserved in *Ifnar1⁻/⁻* mice, but lost in *Casp1⁻/⁻* mice, consistent with the fact that formation of inflammasomes results in caspase-1 activation followed by Il1b release. Inflammasome formation in mice requires priming via NF-κB to induce expression of the scaffold protein NLRP3 (*Bauernfeind et al., 2009*; *Broz and Dixit, 2016*). This priming is reduced in *Casp1⁻/⁻* mice due to the reduction of Il1b cleavage and release and, hence, less NF-κB activation. In addition, cleavage of pro-Il1b was activated in STING ki mice as compared to STING WT (*Figure 6C*, quantified in E, and *Figure 6—figure supplement 1F*). This difference was preserved in *Ifnar1⁻/⁻* mice, but lost in *Casp1⁻/⁻* mice, consistent with the fact that pro-Il1b cleavage is mediated by caspase-1.

To confirm that inflammasomes are activated in STING ki mice, we stained for the inflammasome scaffold apoptosis-associated speck-like protein containing a CARD (ASC). ASC-positive puncta were observed in Iba1-positive and in GFAP-positive cells of striatal slices (*Figure 6F*). Both the area fraction of ASC staining (*Figure 6G*) and the number of puncta within Iba1-positive cells (*Figure 6H*) was higher in slices of adult STING ki mice than in STING WT. ASC-positive dots were also observed in GFAP-positive cells, but not as frequently as in Iba1-positive cells (*Figure 6I* and *Figure 6—figure supplement 1I*). This is consistent with the higher mean intensity of the total ASC signal observed in Iba1-positive cells compared to GFAP-positive cells (*Figure 6—figure supplement 1J*). Inflammasome activation, as reported by ASC staining, was preserved in *Ifnar1⁻/⁻* mice but absent in *Casp1⁻/⁻* mice (*Figure 6G–I*), consistent with the findings obtained using immunoblots (*Figure 6C–E*).

## Type I IFN and inflammasome signaling contribute to neuroinflammation in STING ki mice

To further explore the interaction between inflammatory pathways for neuroinflammation in our STING ki mice, we measured the expression of ISGs, microglia-polarization related genes and concentrations of 13 cytokines and chemokines (summarized in *Figure 7A*). For practical reasons, ISG expression was measured in cortical lysates (*Figure 7—figure supplement 1*) whereas cytokines and genes related to microglia polarisation were measured in striatal lysates (*Figure 7—figure supplement 2*). The N153S STING-induced increase in the interferon dependent genes *Ifi44* and *Mx1* was abrogated in *Ifnar1⁻/⁻* mice, but unaltered in *Casp1⁻/⁻* mice (*Figure 7—figure supplement 1A, B*), consistent with the dependence of *Ifi44* and *Mx1* on type I interferon signalling. Conversely, the N153S STING-induced increase in expression of *Il1b* was reduced in *Casp1⁻/⁻* mice, but not in *Ifnar1⁻/⁻* mice (*Figure 7—figure supplement 1C*), consistent with the fact that *Il1b* expression is inflammasome-dependent. The N153S STING-induced increase in *Cxcl10* expression was present on all backgrounds (*Figure 7—figure supplement 1D*). Expression of *Tnfa* and *Sting1* was not different between STING ki and STING WT in adult mice (*Figure 7—figure supplement 1E, F*).

Activated microglia can exert a continuum of phenotypes between the pro-inflammatory 'M1' phenotype and an 'M2' phenotype associated with trophic factors and tissue repair (*Keren-Shaul et al., 2017*; *Orihuela et al., 2016*). We therefore characterized the effect of STING ki on microglia activation – with and without depletion of *Ifnar1* or *Casp1* – in striatal lysates (*Figure 7—figure supplement 2*). For most cytokines, the concentration was higher in lysates of STING ki mice (*Figure 7A*). When analysed individually, the N153S STING-induced increase was more pronounced in *Ifnar1⁻/⁻* mice, notably for the 'M1' markers IFNα, IFNγ, Cxcl9, Cxcl10, CCL3, and CCL2 (*Figure 7—figure supplement 2A-F*). The concentration of TNFα protein was unaltered (*Figure 7—figure supplement 2G*), consistent with the unaltered gene expression (*Figure 7—figure supplement 1E*). Also, we did not observe a significant difference between STING ki and STING WT for IL-6, CCL4, Il4, IL-10, GM-CSF and VEGF (*Figure 7—figure supplement 2H-M*). As for gene expression in the striatum, on the *Ifnar1⁻/⁻* background, both the 'M1' marker *Nos2* and the 'M2' marker *Ym-1* were higher in STING ki mice than in STING WT (*Figure 7—figure supplement 2N, O*). On WT background and in *Casp1⁻/⁻* mice,

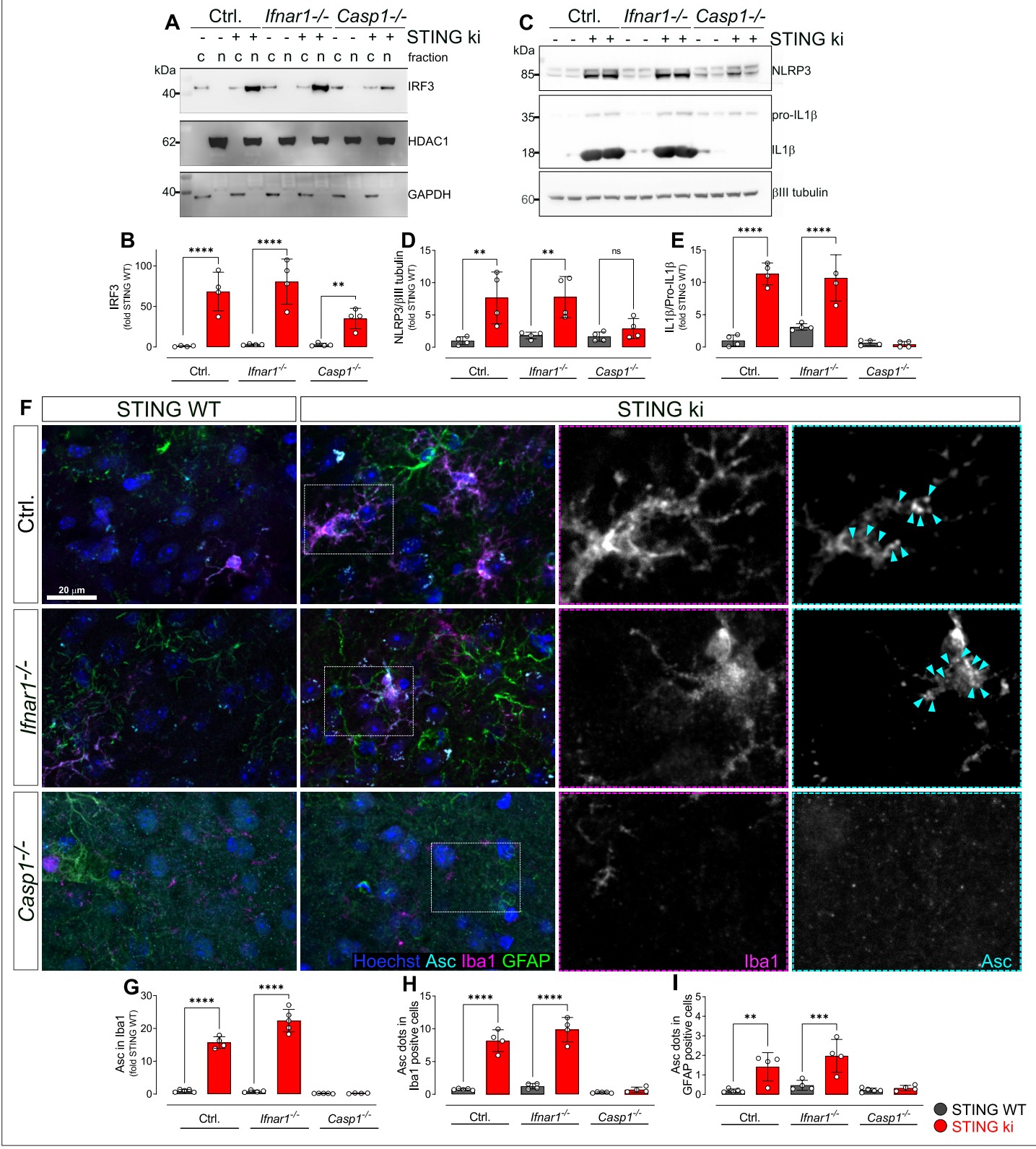

**Figure 6.** Activation of IRF3 and NF-kB-related signalling pathways in double transgenic mice with STING N153S/WT ki and knock-out for Ifnar1 or Caspase-1. (**A**) Representative western blot images showing interferon regulatory factor 3 (IRF3, upper panel), the nuclear marker histone deacetylase 1 (HDAC1, middle panel) and cytoplasmic marker glycerinaldehyd-3-phosphat-dehydrogenase (GAPDH, lower panel) detected from the striatum. Images of the whole membrane stained for the different proteins are shown on **Figure 6—figure supplement 1A-C**. (**B**) Ratio of IRF3 and HDAC1, expressed as

*Figure 6 continued on next page*

*Figure 6 continued*

relative to the mean of STING WT (****: p<0.00001; **: p=0.0043; two-way ANOVA with Tukey post-hoc test, n=4). Ratio of IRF3 and GAPDH expressed as relative to the mean of STING WT is shown on *Figure 6—figure supplement 1D*. (**C**) Representative western blot images showing NLR family pyrin domain containing 3 (NLRP3, upper panel), Il1b and pro-Il1b (middle panel) and the loading control βIII tubulin (lower panel) detected from the striatum. Images of the whole membrane stained for the different proteins are shown on *Figure 6—figure supplement 1E-H*. (**D**) Ratio of NLRP3 and βIII tubulin, expressed as relative to the mean of STING WT (**: p<0.0057; **: p=0.0029; two-way ANOVA with Tukey post-hoc test, n=4). (**E**) Ratio of Il1b and pro-Il1b, expressed as relative to the mean of STING WT (****: p<0.00001; two-way ANOVA with Tukey post-hoc test, n=4). (**F**) Representative images of striatal sections stained for the astroglia marker GFAP (green), microglia marker Iba1 (magenta), apoptosis-associated speck-like protein (ASC, cyan) and Hoechst (blue) from STING WT or STING ki mice on a background of interferon a receptor knockout (*Ifnar1*$^{-/-}$), caspase-1 knockout (*Casp1*$^{-/-}$) or *Ifnar1*$^{+/+}$, *Casp1*$^{+/+}$ (Ctrl.). Scale bar: 20 µm. Magnified insets show Iba1 and ASC, cyan arrowheads indicate ASC specks. (**G**) Area fraction of ASC signal within microglia as relative to the mean of STING WT (****: p<0.00001; two-way ANOVA with Tukey post-hoc test, n=4–5). (**H**) Number of ASC-positive dots within microglia as relative to the mean of STING WT (****: p<0.00001; two-way ANOVA with Tukey post-hoc test, n=4–5). (**I**) Number of ASC-positive dots within astroglia as relative to the mean of STING WT (**: p=0.0083; ***: p=0.0006; two-way ANOVA with Tukey post-hoc test, n=4–5). Graph showing area fraction of ASC signal within astroglia is on *Figure 6—figure supplement 1I*.

The online version of this article includes the following source data and figure supplement(s) for figure 6:

**Figure supplement 1.** Uncropped membranes showing the levels of inflammatory proteins in the striatum.

**Figure supplement 1—source data 1.** Individual, uncropped membranes.

the differences between STING ki and STING WT were not statistically significant, similar to the observations with the protein levels of cytokines. Expression of the 'M2' markers Il4 and *Retnlb* was not altered (*Figure 7—figure supplement 2P, Q*).

A principal component analysis of all measured parameters (*Figure 7—figure supplement 2R*) shows STING ki as the main contributor of the first component and the WT background as the main contributor of the second component. A heatmap of z-transformed values for all parameters (*Figure 7A*) shows a marked difference between STING ki and STING WT animals on WT background and in *Ifnar1*$^{-/-}$ mice but not in *Casp1*$^{-/-}$ mice.

Collectively, these findings suggest a more prominent role for the caspase-1 dependent inflammasome pathway for the N153S STING-dependent activation of the inflammatory mediators we measured. In order to determine functional significance for neuroinflammation, we next measured glial activation in *Ifnar1*$^{-/-}$ and *Casp1*$^{-/-}$ mice (*Figure 7B and C*).

For microglia, N153S STING-induced activation of microglia was still observed in adult *Ifnar1*$^{-/-}$ and *Casp1*$^{-/-}$ mice (*Figure 7B and D*, fold increase and p values in legend). In order to determine whether the extent of N153S STING-induced microglia activation was significantly different between *Ifnar1*$^{-/-}$ mice and Ctrl., we calculated the interaction between the factors 'STING genotype' and '*Ifnar1* genotype' in two-way ANOVA. The extent of microglia activation was significantly smaller in adult *Ifnar1*$^{-/-}$ mice (p=0.004724) than in Ctrl. Similarly, the extent of N153S STING-induced microglia activation was significantly smaller in *Casp1*$^{-/-}$ mice (p=0.000021). Furthermore, the extent of N153S STING-induced microglia activation did not differ between *Ifnar1*$^{-/-}$ and *Casp1*$^{-/-}$ (p=0.0805731). N153S STING-induced microglia activation therefore depends both on type 1 IFN signalling and on inflammasome signalling.

For astroglia (*Figure 7C and E*), the extent of N153S STING-induced activation was significantly reduced from 26-fold in Ctrl. to 16-fold in *Ifnar1*$^{-/-}$ (p=0.0158378 for interaction of twoway ANOVA) and to fourfold in *Casp1*$^{-/-}$ (p<0.00001 for interaction of twoway ANOVA). In fact, no statistically significant N153S STING-induced activation of astroglia was observed on the *Casp1*$^{-/-}$ background. These findings suggest that astroglia activation might depend more on inflammasome signalling than on type I IFN-dependent signalling.

## Oxidative stress in STING ki mice

Neuroinflammation can be associated with oxidative stress, and dopaminergic neurons are particularly sensitive to oxidative stress. We therefore measured reactive oxygen species (ROS) in lysates obtained from the striatum of adult mice. Mitochondrial ROS levels were higher in STING ki than in STING WT (*Figure 8A*). Similarly, cytoplasmic ROS levels were increased in STING ki as compared to STING WT (*Figure 8B*). The N153S STING-induced increase was also observed on the *Ifnar1*$^{-/-}$ background, but not on the *Casp1*$^{-/-}$ background. To report the activity of the (inducible) nitric oxide synthase, we measured the concentration of nitrite in the striatal lysates (*Figure 8C*). As for ROS, the

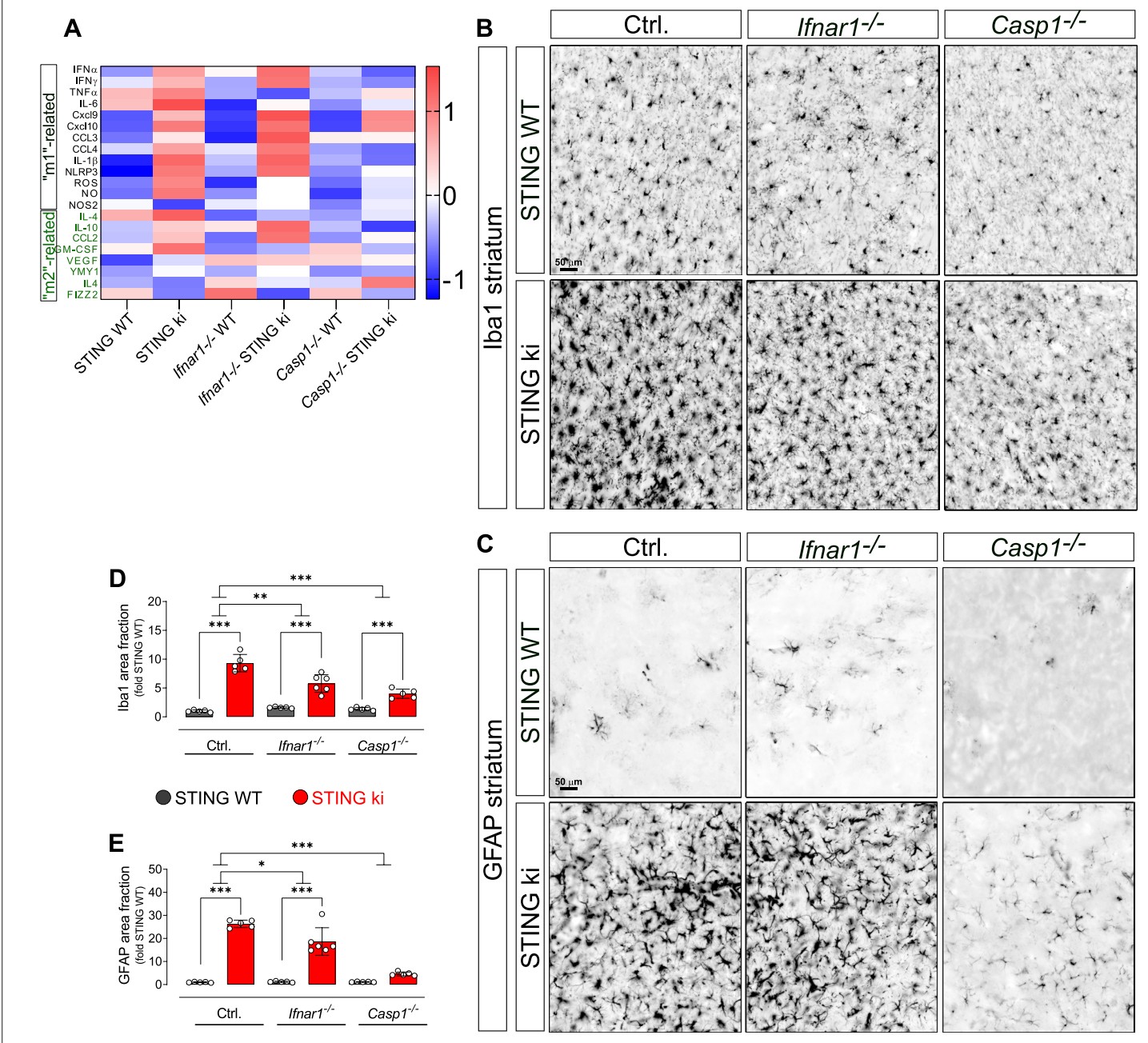

**Figure 7.** Neuroinflammation in adult double transgenic mice with STING ki and knock-out for Ifnar1 or Caspase-1. (**A**) Heatmap showing z-scores of different inflammatory markers in double transgenic mice with STING N153S/WT ki and knock-out for Ifnar1 or Caspase-1. Graphs with data for each mediator individually are on (*Figure 7—figure supplement 2*). (**B**) Representative images of striatal sections stained for the microglia marker Iba1. Sections were obtained from adult STING WT (upper images) or STING ki (lower images) mice on a background of interferon a receptor knockout (*Ifnar1⁻/⁻*), caspase-1 knockout (*Casp1⁻/⁻*) or *Ifnar1⁺/⁺*, *Casp1⁺/⁺* (Ctrl.). Scale bar: 50 μm. (**C**) Representative images of striatal sections stained for the astroglia marker GFAP from STING WT (upper images) or STING ki (lower images) mice on a background of interferon a receptor knockout (*Ifnar1⁻/⁻*), caspase-1 knockout (*Casp1⁻/⁻*) or *Ifnar1⁺/⁺*, *Casp1⁺/⁺* (Ctrl.). Scale bar: 50 μm. (**D**) Area fraction positive for Iba1, normalized to the mean of STING WT brains (differences in ⁺/⁺ mice ***: p=0.0000001; for *Ifnar1⁻/⁻* ***: p=0.000003; for *Casp1⁻/⁻* ***: p=0.0029374; two-way ANOVA with Bonferroni post-hoc test, n=5–6). (**E**) Area fraction positive for GFAP, normalized to STING WT on Ctrl. Background (***: p=0.0000 for STING WT vs STING ki on Ctrl.; ***: p=0.0000 on *Ifnar1⁻/⁻*; background, ***: p=0.0006 on Casp1-/- background; two-way ANOVA with Bonferroni post-hoc test, n=5–6).

The online version of this article includes the following figure supplement(s) for figure 7:

**Figure supplement 1.** Expression of IFN- and NF- κ B/inflammasome related genes in the cortex of double transgenic mice with STING N153S/WT ki and knock-out for Ifnar1 or Caspase-1.

*Figure 7 continued on next page*

*Figure 7 continued*

**Figure supplement 2.** Levels of inflammatory mediators and gene expression in the striatum of double transgenic mice with STING N153S/WT ki and knock-out for Ifnar1 or Caspase-1.

nitrite concentration was higher in lysates of STING ki mice than in STING WT; this difference was also observed on *Ifnar1*[-/-] and *Casp1*[-/-] background.

Taken together, these findings demonstrate increased oxidative stress in the brain of STING ki mice, which constitutes a potential mechanism by which neuroinflammation might contribute to the degeneration of dopaminergic neurons (***Figure 1***).

### Type I IFN and inflammasome signalling contribute to neurodegeneration in STING ki mice

Overall, several parameters of neuroinflammation were blunted in *Casp 1*[-/-] mice, but not in *Ifnar1*[-/-] mice. We wanted to use this difference to examine the question whether activation of type I IFNs or inflammasomes is more important for degeneration of dopaminergic neurons (***Figure 9***.). Unfortunately, however, the density of dopaminergic axon terminals in the striatum was already lower in *Ifnar1*[-/-] mice and *Casp 1*[-/-] mice with WT STING than in control *Ifnar1*[+/+] and *Casp1*[+/+] mice (p=0.0143071 and p=0.0000248, two-way ANOVA). These findings suggest that *Ifnar1* and *Casp1* are required for proper proliferation, maturation and/or maintenance of dopaminergic neurons and their axon terminals. Indeed, the Wnt-β-catenin pathway is regulated by interferons (***Kovács et al., 2019***) and regulates the differentiation of midbrain dopaminergic neurons (***Szegő et al., 2017***). Furthermore, Il1b induces the differentiation of dopaminergic neurons (***Ling et al., 1998***; ***Rodriguez-Pallares et al., 2005***). We found marginally reduced *Il1b* expression in adult *Casp1*[-/-] mice (***Figure 7—figure supplement 1C***, STING WT), which could explain reduced density of dopaminergic fibers observed in *Casp1*[-/-] mice.

Both in *Ifnar1*[-/-] and in *Casp 1*[-/-] mice, the N153S STING-induced degeneration of dopaminergic axon terminals in the striatum was less pronounced than in the control *Ifnar1*[+/+-] and *Casp1*[+/+] mice (***Figure 9A and B***; p=0.00157 and p=0.007326 for interaction in two-way ANOVA). N153S STING-induced fibre loss was not statistically different between *Ifnar1*[-/-] and *Casp1*[-/-] mice (p=0.468 for interaction). Consistent with the less pronounced degeneration of dopaminergic axon terminals, the N153S STING-induced reduction in striatal dopamine was not observed in *Ifnar1*[-/-] and *Casp1*[-/-] mice (***Figure 9C*** and ***Figure 9—figure supplement 1***). These findings suggest that both pathways contribute to the degeneration of dopaminergic axon terminals, and blocking either pathway could reduce neurodegeneration.

### Discussion

In this work, we demonstrated that the expression of the constitutively active STING variant N153S causes neuroinflammation, which is followed by degeneration of dopaminergic neurons and aSyn

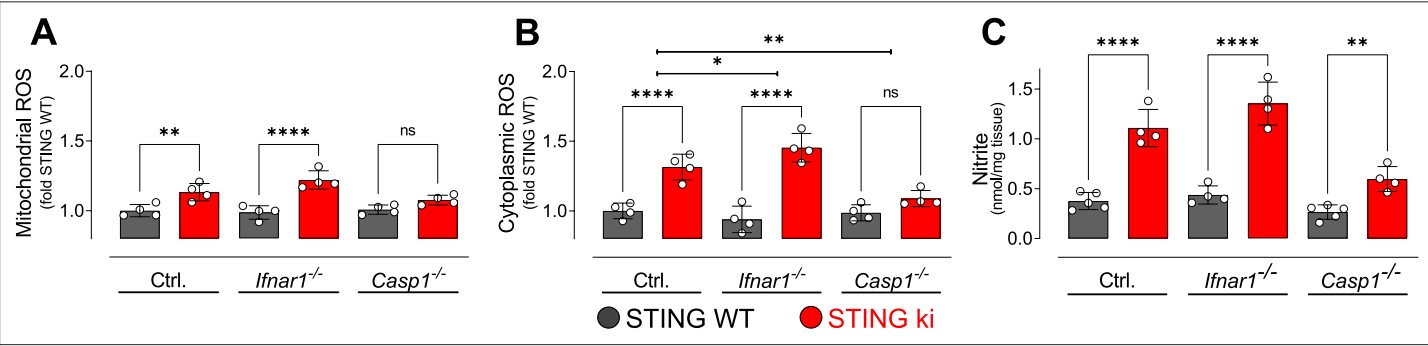

**Figure 8.** Oxidative stress in the striatum of double transgenic mice with STING N153S/WT ki and knock-out for Ifnar1 or Caspase-1. (**A**) Mitochondrial reactive oxygen species (ROS) of striatal lysates determined using MitoSOX and expressed relative to the mean of STING WT (\*\*: p=0.0072; \*\*\*\*: p<0.00001; two-way ANOVA with Tukey post-hoc test, n=4). (**B**) Cytoplasmic ROS of striatal lysates determined using DCFH-DA and expressed relative to the mean of STING WT (\*\*\*\*: p<0.00001; \*: p=0,034; \*\*: p=0.0069; two-way ANOVA with Tukey post-hoc test, n=4). (**C**) Nitrite levels as markers of nitric oxide activity measured in striatal lysates (\*\*\*\*: p<0.00001; \*\*: p=0.0052; two-way ANOVA with Tukey post-hoc test, n=4–5).

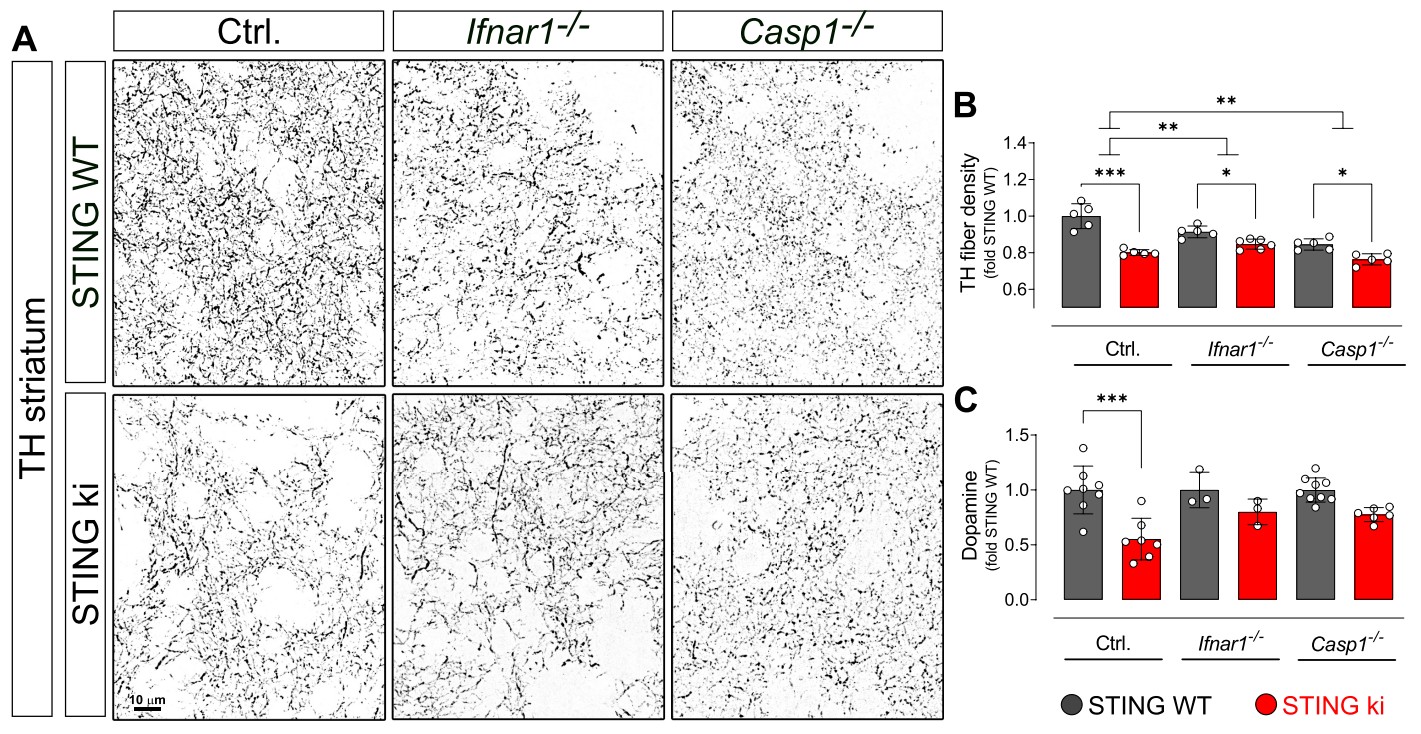

**Figure 9.** Degeneration of dopaminergic neurons in double transgenic mice with STING N153S/WT ki and knock-out for Ifnar1 or Caspase-1. (**A**) Representative images of striatal sections stained for tyrosine hydroxylase (TH) from STING WT (upper images) or STING ki (lower images) mice on a background of interferon a receptor knockout (*Ifnar1*⁻/⁻), caspase-1 knockout (*Casp1*⁻/⁻) or *Ifnar1*⁺/⁺, *Casp1*⁺/⁺ (Ctrl.). Scale bar: 10 μm. (**B**) Area fraction positive for TH, normalized to STING WT on Ctrl. background (***: p=0.0000 for Ctrl. background; *: p=0.043 for *Ifnar1*⁻/⁻; **: p=0.0126845 for *Casp1*⁻/⁻; for interaction between Ctrl. background and *Ifnar1*⁻/⁻: p=0.00157; between Ctrl. and *Casp1*⁻/⁻: p=0.007326; two-way ANOVA with Bonferroni post-hoc test, n=5–6). (**C**) Concentration of dopamine in striatal lysates of STING WT and STING ki mice, normalized to STING WT on Ctrl. background. (***: p=0.0005; t-test, n=5–6). Dopamine metabolism is shown on *Figure 9—figure supplement 1*.

The online version of this article includes the following figure supplement(s) for figure 9:

**Figure supplement 1.** Dopamine metabolism in the striatum of double transgenic mice with STING N153S/WT ki and knock-out for Ifnar1 or Caspase-1.

pathology. N153S STING-induced microglia activation and degeneration of dopaminergic neurons involve type I IFN and inflammasome-mediated signalling; astroglia activation might depend predominantly on the inflammasome pathway.

## Constitutive STING activation causes neuroinflammation

STING is expressed in the central nervous system. Its expression is highest in microglia, but it can also be detected in astroglia and neurons (*Jeffries and Marriott, 2017*). Microglia are the primary immune cells of the central nervous system (*Wolf et al., 2017*) and highly activated upon pathogen invasion or tissue damage. Microglia activation in juvenile (*Figure 2*) and adult (*Figure 1*) STING ki mice is therefore in line with the increased inflammatory phenotype found in the lungs and spleen of STING ki mice (*Luksch et al., 2019*; *Siedel et al., 2020*) and in patients with STING-associated vasculopathy with onset in infancy syndrome (*Liu et al., 2014*). SAVI patients may suffer from various neurological symptoms, including calcification of basal ganglia, but neurological symptoms are not a core feature of the human SAVI disease (*Frémond et al., 2021*).

Consistent with this morphologically defined neuroinflammatory phenotype, we observed increased expression of the ISGs *Ifi44* and *Mx1* in striatum and cortex (*Figure 4G and H* and supplement 1A, B). Moreover, we observed nuclear translocation of pSTAT3, which was already present in juvenile mice, but more prominent in adult mice (*Figure 5C*), and nuclear translocation of IRF3 (*Figure 6A and B*). These effects are associated with IFN signalling, the main pathway downstream of STING (*Decout et al., 2021*), and demonstrate that expressing the N153S mutant of STING indeed activates

IFN-dependent pathways in these brain regions. Accordingly, microglia activation was reduced when N153S STING was expressed in mice deficient for *Ifnar1* (*Ifnar1*^-/-^, *Figure 7D*), confirming that the type I IFN pathway is involved in STING-dependent microglia activation - as previously demonstrated by others (*Warner et al., 2017*). However, microglia activation by N153S STING was not completely blocked in *Ifnar1*^-/-^ mice, suggesting that additional pathways are involved.

Indeed, next to type I IFN signalling, we also observed nuclear translocation of NF-κB (*Figure 5D*), which was already prominent in juvenile mice, and suggests activation of NF-κB-dependend signalling in STING ki mice. Additionally, we observed increased expression of the NF-κB target genes *Il1b* (*Hiscott et al., 1993*) and *Tnfa* (*Collart et al., 1990*). Expression of *Il1b* was significantly increased in lysates of SN (*Figure 4E*), striatum (*Figure 4K*) and cortex (*Figure 4—figure supplement 1F*). For *Tnfa*, we observed a significant increase in juvenile STING ki (*Figure 4—figure supplement 1E*), but no significant change in adult mice (*Figure 4D and J*). Activation of the NF-κB pathway by STING is consistent with previous work by others (*Yum et al., 2021*). Furthermore, increased expression of *Il1b* and *Tnfa* in a mouse model of TDP-43 proteinopathy was recently found to be STING dependent (*Yu et al., 2020*).

One of the key signalling events downstream of STING and NF-κB is the activation of inflammasomes. Formation and activation of inflammasomes in STING ki mice was observed morphologically by quantification of ASC puncta (*Figure 6F–I*) and biochemically through Il1b cleavage (*Figure 6C and E*) and accumulation of NLRP3 (*Figure 6C and D*). Accordingly, the inflammasome dependent cleavage of Il1b was absent in STING ki mice with *Casp1* depletion (*Figure 6C and F*). The increased abundance of NLRP3 in STING ki mice was not observed in *Casp1*^-/-^ mice (*Figure 6D*), suggesting that this regulation, too, results from inflammasome activation and not directly from NF-κB signalling. Inflammasome activation is consistent with previous work about systemic inflammation (*Luksch et al., 2019*). The extent of microglia activation by N153S STING was reduced in *Casp1*^-/-^ mice (*Figure 7D*). Therefore, our results suggest that STING-dependent microglia activation involves inflammasome activation, in addition to type I IFN-dependent pathways. Activation of astroglia, in contrast, was completely blocked in *Casp1*^-/-^ mice (*Figure 7E*), suggesting that STING-induced astroglia activation might depend mainly on inflammasome signalling. Mechanistically, astroglia activation could result directly from the expression of N153S STING in astroglia, but also indirectly through the activation of microglia (*Kwon and Koh, 2020*; *Wolf et al., 2017*). The differential effect in *Ifnar1*^-/-^ and *Casp1*^-/-^ mice suggests, however, that astroglia activation is not only a downstream consequence of microglia activation.

## Chronic STING activation leads to the degeneration of dopaminergic neurons

We observed a reduced number of TH-positive neurons in the substantia nigra of STING ki mice, a reduced density of dopaminergic axon terminals and a reduced concentration of striatal dopamine (*Figure 1G–I*). These changes occurred in adult mice and were not present in juvenile mice (*Figure 2G–I*). The degeneration of dopaminergic neurons thus is a consequence of the prolonged inflammatory changes in our model.

On the *Ifnar1*^-/-^ and *Casp1*^-/-^ backgrounds, the density of dopaminergic axon terminals was already reduced (*Figure 9B*), making it difficult to assess the effect of N153S STING. Nonetheless, the relative reduction in TH fibre density in STING ki mice, as compared to STING WT mice, was smaller on the *Ifnar1*^-/-^ and *Casp1*^-/-^ backgrounds than in controls (*Figure 9B*), as was the extent of dopamine depletion (*Figure 9C*). These findings are consistent with the partial rescue of microglia activation in *Ifnar1*^-/-^ and *Casp1*^-/-^ mice (*Figure 7D and E*). They suggest that glia activation and secretion of inflammatory cytokines contribute to the degeneration of dopaminergic neurons – in line with previous findings demonstrating a role for inflammation in the pathogenesis of PD (*Hall et al., 2018*; *Mollenhauer et al., 2019*). In further studies, the developmental effects of *Ifnar1* and *Casp1* deficiency could be circumvented by using conditional knockout mice or pharmacological inhibitors. Oxidative stress was increased in STING ki mice (*Figure 8*) and cytoplasmic ROS were partially reduced in *Casp1*^-/-^ mice (*Figure 8B*), suggesting that oxidative stress might contribute to the N153S STING-induced degeneration of dopaminergic neurons. In addition, constant STING activity within dopaminergic neurons could contribute to their degeneration – both in our STING ki mice and in the pathogenesis of PD. Indeed, dopaminergic neurons accumulate oxidative damage as a consequence of dopamine

synthesis and electrical pacemaking activity (*Guzman et al., 2010*), and even moderate oxidative stress can stimulate the STING pathway (*Sliter et al., 2018*; *West et al., 2015*).

Secreted immune mediators may trigger neuron loss upon STING activation (*Murthy et al., 2020*). For instance, TNFR1 is a member of the death receptor family, and can induce the recruitment of TNFR1-associated death domain protein (TRADD) and apoptosis upon Tnfa binding (*Jayaraman et al., 2021*; *Murthy et al., 2020*). Indeed, inhibition of TRADD can restore proteostasis and inhibits mutant tau-induced cell death (*Xu et al., 2020*). Next to Il1b cleavage, caspase-1 might also induce pyroptosis, an inflammatory form of cell death. After cleavage by caspase-1, gasdermin-D forms pores in the cell membrane (*Shi et al., 2017*). Rupture of the plasma membrane leads to the release of cytoplasmic content into the extracellular space, further fuelling inflammatory response (*Voet et al., 2019*). Furthermore, loss of ion gradient across the membrane makes neuronal functioning impossible. Besides increased oxidative stress or secreted immune mediators, neuronal STING activation may induce directly apoptosis. Emerging evidence suggests that STING activation induces endoplasmic reticulum (ER) stress and disrupt calcium homeostasis in various cell types (*Cui et al., 2016*; *Petrasek et al., 2013*; *Wu et al., 2019*), that, in turn, result in the loss of dopaminergic neurons (*Gong et al., 2019*). Further work with cell-type specific N153S STING expression will be required to determine the importance of STING activation within dopaminergic neurons for their degeneration in this model.

The degeneration of dopaminergic neurons observed in these mice was not severe. Based on findings with other mouse models of PD by ourselves and others (*Bentea et al., 2015*; *Gong et al., 2019*; *Oliveras-Salvá et al., 2013*), we do not expect a dopamine-dependent motor phenotype in these mice. Moreover, motor assessment is complicated by systemic inflammation in STING ki mice, affecting the lung and further organs (*Luksch et al., 2019*).

## Accumulation of protein aggregates in STING ki mice

We observed an accumulation of phosphorylated and Triton X-100-insoluble aSyn, and an increased number of Thioflavin S positive cells in STING ki mice (*Figure 3A–E*). Collectively, these findings suggest that chronic STING activation induces aSyn pathology. They are consistent with the recent observation that priming rats with a mimic of viral dsDNA precipitates aSyn pathology (*Olsen et al., 2019*) and with the aSyn aggregation following viral encephalitis (*Bantle et al., 2021*). Inflammatory signals are therefore active promotors of aSyn pathology and not only responsive to aSyn pathology. Thioflavin S staining is, however, not specific for aSyn pathology, and STING-induced neuroinflammation could involve further proteins. Indeed, activation of NF-κB signalling in microglia was recently demonstrated to drive tau pathology (*Wang et al., 2022*).

The mechanism by which prolonged neuroinflammation leads to aSyn pathology is still unknown. Both inflammation and IFN induce the expression of the double-stranded (ds) RNA-dependent protein kinase (PKR) (*Gal-Ben-Ari et al., 2018*). PKR can phosphorylate aSyn at serine 129, resulting in aSyn pathology (*Reimer et al., 2018*). Moreover, clearance of aSyn aggregates occurs primarily through autophagy (*Ebrahimi-Fakhari et al., 2011*), and INFα increases expression of mammalian target of rapamycin (mTOR) (*Liu et al., 2016*), which is expected to reduce autophagy initiation. On the other hand, acute STING activation can induce autophagy (*Hopfner and Hornung, 2020*; *Liu et al., 2018*; *Moretti et al., 2017*). aSyn pathology in our model therefore could be explained by an exhaustion of the autophagy machinery, as it was suggested recently (*Bido et al., 2021*), and the overall effect of inflammatory pathways on autophagy and aSyn pathology could be bimodal. Accordingly, degeneration of dopaminergic neurons and accumulation of aSyn aggregates was also observed in mice deficient for IFNβ (*Ejlerskov et al., 2015*; *Magalhaes et al., 2021*).

## Conclusions

This work provides evidence in a genetic model that chronic activation of STING signalling is sufficient for degeneration dopaminergic neurons. It has been demonstrated previously that blocking STING signalling or depleting STING is beneficial in animal models of ALS (*Yu et al., 2020*) and PD (*Sliter et al., 2018*). Yet, neuroinflammatory signalling involves many positive and negative feedback loops, so studies activating and blocking specific nodes are important to advance understanding and identify treatment targets. The interconnected nature and redundancy of neuroinflammation signalling was highlighted, for instance, by the strong inhibition of inflammasome signalling in $Casp1^{-/-}$ mice, but the incomplete rescue of microglia activation and neurodegeneration.

This work is preliminary in many aspects. We have focused on dopaminergic neurons because we have been interested in pathways that cause their degeneration in the past, and STING has been linked to PD pathogenesis (*Sliter et al., 2018*). Yet, STING-induced neurodegeneration is expected to affect further neuron populations, which need to be assessed in future studies. Indeed, STING-induced neuroinflammation was not restricted to the substantia nigra, but also observed, for instance, in the cortex. Also, Thioflavin S accumulation likely involves other proteins than aSyn. STING-induced neurodegeneration and proteinopathies therefore need to be assessed more systematically in future studies, addressing the question whether some brain regions and neuron populations are more susceptible to STING-induced damage than others. Finally, genetic activation of STING in cell-types will be required to delineate the contributions of STING activation within microglia and neurons.

## Materials and methods

Source of chemicals, antibodies, composition of buffers, equipment and software used in this study are listed in the Key Resources Table.

### Animals

All animal experiments were carried out in accordance with the European Communities Council Directive of November 24, 1986 (86/609/EEC) and approved by the Landesdirektion Dresden, Germany. Mice of both sexes were housed under a 12 hr light and dark cycle with free access to pelleted food and tap water in the Experimental Center, Technische Universität, Dresden, Germany. Heterozygous STING N153S/WT ki mice (STING ki) were previously described (*Luksch et al., 2019*). STING ki or STING WT mice were crossed to *Ifnar1*$^{-/-}$ mice (a gift from Axel Roers, Dresden, Germany; *Siedel et al., 2020*) and *Casp1*$^{-/-}$ mice (a gift from Stefan Winkler, Dresden, Germany ;*Reinke et al., 2020*).

For all genotypes, five-week-old (from here referred as juvenile) or 20–23 week-old (referred as adult) animals were sacrificed with an overdose of isoflurane (Baxter, Lessines, Belgium). For analyses of proteins, gene expression, catecholamine and oxidative stress, brains were rapidly removed from the skull and washed in ice-cold Tris-buffered saline (TBS, pH 7.4). Cortex and striatum were dissected, snap-frozen in liquid nitrogen and stored at –80 °C until use. For histology, mice were perfused transcardially with 4% paraformaldehyde (PFA) in TBS. After post-fixation (4% PFA, overnight) and cryoprotection (30% sucrose in TBS), 30 µm-thick coronal brain sections were cut in a cryostat (Leica, Germany).

### Immunofluorescence stainings of mouse brain sections

To quantify the number of dopaminergic neurons in the substantia nigra (SN), every fifth section throughout the entire SN was stained for tyrosine hydroxylase (TH) as previously (*Szegő et al., 2021*). In brief, after blocking (2% bovine serum albumin, 0,3% Triton X-100 in TBS; 1 hr RT), sections were incubated with the primary antibody in blocking solution (two overnights), followed by the fluorescently labelled secondary antibody (Alexa 488 conjugated donkey anti-sheep, 1:2000, overnight). Sections were counterstained with Hoechst and mounted with Fluoromount-G.

To quantify the density of dopaminergic axon terminals (fibers) and neuroinflammation in the striatum, every sixth section throughout the entire striatum was stained with a cocktail of primary antibodies: TH (Pel Freeze, P40101, 1:1000), Iba1 (Wako, 019–19741, 1:1000) and GFAP (abcam, ab4674, 1:2000). As fluorescently labelled secondary antibodies, Alexa 488 conjugated donkey anti-sheep, Alexa 555 conjugated donkey anti-rabbit, Alexa 647 conjugated donkey anti-chicken were used (1:2000, overnight). Sections were counterstained with Hoechst and mounted with Fluoromount-G.

To detect cells containing inclusions of β-sheet containing proteins, mounted brain sections were stained with 0.1% Thioflavin-S in 70% ethanol 25 min, rinsed in 80% ethanol for 5 min and in distilled water for 5 min. After staining the nuclei with NucSpot nuclear stain, sections were mounted in Fluoromount-G (*Szegő et al., 2019*).

To quantify ASC-inflammasome assembly in glia cells, every tenth section throughout the entire striatum was stained with a cocktail of the following primary antibodies: ASC (Adipogen, 1:700, AG-25B-0006-C100), Iba1 (Synaptic Systems, 234004, 1:1000), and GFAP (abcam, ab4674, 1:2000). As fluorescently labelled secondary antibodies, Alexa 488 conjugated donkey anti-rabbit, Alexa 555

conjugated goat anti-guinea pig and Alexa 647 conjugated donkey anti-chicken were used (1:2000, overnight). Sections were counterstained with Hoechst and mounted with Fluoromount-G.

## Quantification of dopaminergic neuron number, striatal fiber density and gliosis

The number of dopaminergic somata in the SN was determined by supervised manual counting by an experienced investigator blinded to the experimental groups. For each animal, every fifth section throughout the rostro-caudal extent of the SN (2.54 to −3.88 mm posterior to Bregma based on *Paxinos and Franklin, 2001*) was incorporated into the counting procedure. In each section, z stacks were acquired (step size: 2 µm, 5 slices in total) from both hemispheres with a 20 x objective (N.A 0.8, Axio Imager 2, Zeiss). Stacks were stitched to reconstruct the entire SN. After adjusting the threshold and carefully marking the borders of the SN, only TH-positive cell bodies with a visible nucleus in the blue channel were manually counted by ImageJ (1,53 c; Cell Counter plugin). All TH-positive neurons of each slices were counted. The total number of neurons for the entire SN of that hemisphere was estimated by multiplying the counted cell number by five, since every fifth section was used for this analysis.

For quantification of gliosis, five fluorescent images were acquired from every sixth striatal section stained for GFAP and Iba1 using a 20 x objective. After adjusting the threshold and noise removal (Background subtraction, rolling ball radius 50) from the individual images (separately for GFAP and Iba1 channels), the area fraction was determined by ImageJ from ten regions of interest per image. Results were analyzed using a generalized linear mixed model (glm) in RStudio with a hierarchically nested design (expressed as percent area) as previously (*Szegő et al., 2021*).

From the same striatal sections, the density of the dopaminergic axon terminals (fibers) was determined as described previously (*Szego et al., 2012*). In brief, z-stack images were acquired (five planes, 0.5 µm step size, 100 x objective, N.A. 1.4; Axio Imager 2, Zeiss). TH-positive fibers were delineated from the maximal intensity projection (ImageJ) after adjusting the threshold, noise removal and binarization, and density was expressed as percent area. Every sixth section per animal, five images per section and ten boxes per image were analyzed in a hierarchically nested design as above.

For quantification of the mean fluorescent intensity of the ASC signal, binary masks were first created (see above) separately for microglia (Iba1 channel) and astroglia (GFAP channel). After removing background fluorescence (Background subtraction, rolling ball radius 50) from the individual ASC images, mean fluorescent intensity of the ASC signal was measured by ImageJ from all glia cells from a given microscope image. After adjusting the threshold of the ASC signal on the same individual images, area fraction of the ASC signal and the number of individual dots (ASC-specks) within microglia and within astroglia were also determined separately by ImageJ.

## Protein analyses

To detect protein changes, cortical and striatal tissue were mechanically lysed in a buffer containing 250 mM sucrose, 50 mM TRIS (pH 7,5), 1 mM EDTA, 5 mM $MgCl_2$, 1% Triton X-100 in the presence of protease and phosphatase inhibitors (MedChem Express) as previously (*Szego et al., 2012*). Samples were centrifuged (14,000 g, 30 min, 4 °C) and protein concentration in the supernatant was determined with the BCA method (ThermoFisher, Germany). After boiling with 4 x Laemmli buffer (1 M Tris pH 6.8, 0.8% SDS, 40% glycerol, 5% β-mercaptoethanol, traces of bromophenol blue, 5 min, 95 °C), 5 µg protein was loaded onto a 4–20% Tris/glycine SDS gel for western blot analysis. Membranes were fixed in paraformaldehyde (10 min, RT) and blocked with 1% bovine serum albumin, 0,05% Tween 20 in TBS. Membranes were incubated first in the presence of antibodies against phosphorylated aSyn and aSyn, (Cell Signaling), then with βIII-tubulin as loading control (overnight, 4 °C). Following washing, membranes were incubated in the presence of horseradish peroxidase-conjugated secondary antibodies (donkey anti-mouse or donkey anti-rabbit). Signal was detected using chemiluminescent substrate and a camera-based system. ImageJ was used to determine the optical density of protein bands and all data were analyzed for each group (n = 5 animals/group) based on 3 independent blots. Optical densities were normalized to the expression of the density of the tubulin loading control of the same sample, and then expressed relative to the WT animals.

## Triton X-100 solubility

For detection of aSyn in the Triton X-100 insoluble fraction, snap-frozen substantia nigra and striatal tissue were lysed in PBS containing 1% of Triton X-100, phosphatase inhibitors and protease inhibitors. Lysates were vortexed and incubated for 30 min at 4 °C. After centrifugation (14,000 g, 30 min, 4 °C), supernatant was used as Triton X-100 soluble fraction. The pellet was washed in ice-cold PBS, centrifuged again and re-dissolved with sonication (10 s) in 50 µl buffer containing 2% SDS, 75 mM Tris, 15% glycerin, 3.75 mM EDTA pH 7.4 and protease inhibitors. Solution was briefly centrifuged (5 min, 14 000 g, RT), and it was used as Triton X-100 insoluble fraction. 10 µg of Triton X-100 soluble lysate or 10 µl Triton X-100 insoluble fraction were loaded onto a 4–20% Tris/glycine SDS gel for western blot analysis, and processed as above to detect signals of aSyn and βIII-tubulin. To obtain the ratio of aSyn within the Triton X-100 soluble and insoluble fractions, data was normalized (1) to the tubulin loading control within line, then (2) to the soluble aSyn/tubulin ration, and finally (3) to the values of STING WT animals.

## Biochemical assays

Cytoplasmic reactive oxygen species (ROS) production and mitochondrial ROS production, total nitrite levels were measured as described previously (*Pais et al., 2013*; *Szegő et al., 2019*). In brief, striatal tissue were lysed in ice-cold in PBS and centrifuged (14,000 g, 30 min, 4 °C). Cytoplasmic ROS was measured using dichlorodihydrofluorescein diacetate (DCFH-DA; 10 mM, 37 C, 30 min), mitochondrial ROS was measured using MitoSOX (200 nM, 20 min, 37 C). Fluorescence at 485/530 nm (DCFH-DA) or 510/580 nm (MitoSox) was measured with plate reader. The production of NO by iNOS was measured indirectly by assaying nitrites in the lysate using the Griess reaction (*Pais et al., 2013*). In brief, 10 µg of lysates were incubated with equal amount of Griess reagent (1% sulphanilamide, 0.1% naphthylethylenediamine in 2% phosphoric acid solution) for 20 min at room temperature. Absorbance was read at 550 nm.

## Gene expression analyses

Total RNA was extracted from snap frozen dissected prefrontal cortex tissue by using the RNeasy Mini Kit (Qiagen, Germany) according to the manufacturer's instructions. cDNA was generated by MMLV reverse transcription (Promega Germany). Quantitative Real Time PCR assays were carried out by using QuantStudio 5 (Thermo Fisher Scientific, Germany) and GoTaqaPCR Master Mix with SYBR green fluorescence (Promega, Germany). PCR primer sequences were retrieved from the Primer Bank database (*Spandidos et al., 2010*). Expression of genes was normalized to the expression of the housekeeping genes (*Hprt1*, *Rpl13a*, *Eef2*) and to the STING WT by using the ΔΔCt method. Sequences of primers are listed in the Key Resources Table.

## Legend Plex

LEGENDplex is a multiplex bead-based assay using the basic methodology of ELISA assay. Briefly, snap-frozen dissected STR tissue was homogenized in PBS add 0.5 % NP-40 and protease inhibitor cocktail (Sigma-Aldrich, Germany). The clarified supernatant after centrifugation was used for cytokine determination by Mouse cytokine release syndrome panel (Biolegend) according manufactures introductions. Quantity of cytokines was calculated per mg protein in each lysate.

## Statistical analyses

In graphs, markers represent individual animals; lines represent mean and standard deviation (SD) of all animals. Data normality was tested by the Kolmogorov-Smirnov test and graphically by QQ plot (R, version 2.8.0; *R Development Core Team, 2020*). Grubbs test was used to identify outliers. t-test, Mann-Whitney test or two-way ANOVA were performed using GraphPad Prism (Versions 5.01 and 9.0.0). Linear regression and principal component analysis was performed using R (version 4.2.1.). For generalized linear mixed-effects model (*Szegő et al., 2021*), animal, section and image were used as random effects nested within each other (R package: lme4). p values are indicated in the graphs by symbols with * representing $p<0.05$, ** representing $p<0.01$, *** representing $p<0.001$. Exact p values are given in the Figure legends.

The number of replicates used for the study was determined by prior experience and did not result from a formal sample size estimation. The 'n' indicated for each figure in the figure legend represents the number of animals (biological replicates).

## Acknowledgements

We thank Min Ae Lee-Kirsch (TU Dresden) for very helpful comments on the manuscript and Andrea Kempe, Annett Böhme, Katrin Höhne and Kristin Wogan for excellent technical assistance. Funding This work was supported by the German Research Foundation (DFG) SFB TRR237, B18 and FA-658-3-1.

## Additional information

### Funding

| Funder | Grant reference number | Author |
|---|---|---|
| Deutsche Forschungsgemeinschaft | FA-658-3-1 | Björn H Falkenburger |
| Deutsche Forschungsgemeinschaft | SFB TRR237, B18 | Angela Rösen-Wolff |

The funders had no role in study design, data collection and interpretation, or the decision to submit the work for publication.

### Author contributions

Eva M Szego, Conceptualization, Data curation, Formal analysis, Investigation, Writing – original draft, Validation, Visualization, Writing – review and editing, Methodology; Laura Malz, Nadine Bernhardt, Investigation, Writing – review and editing; Angela Rösen-Wolff, Björn H Falkenburger, Conceptualization, Resources, Supervision, Funding acquisition, Validation, Methodology, Writing – original draft, Project administration; Hella Luksch, Conceptualization, Data curation, Formal analysis, Supervision, Investigation, Methodology, Writing – original draft, Project administration

### Author ORCIDs

Eva M Szego ⓘ http://orcid.org/0000-0002-2629-2131
Nadine Bernhardt ⓘ http://orcid.org/0000-0002-3188-8431
Angela Rösen-Wolff ⓘ http://orcid.org/0000-0002-9613-5879
Björn H Falkenburger ⓘ http://orcid.org/0000-0002-2387-526X
Hella Luksch ⓘ http://orcid.org/0000-0001-7070-4992

### Ethics

All animal experiments were carried out in accordance with the European Communities Council Directive of November 24, 1986 (86/609/EEC) and approved by the Landesdirektion Dresden, Germany.

### Decision letter and Author response

Decision letter https://doi.org/10.7554/eLife.81943.sa1
Author response https://doi.org/10.7554/eLife.81943.sa2

## Additional files

### Supplementary files
- MDAR checklist
- Source data 1. Data used to generate graphs.

### Data availability

All data generated or analysed during this study are included in the manuscript and supporting file.

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

# Appendix 1

## Appendix 1—key resources table

| Reagent type (species) or resource | Designation | Source or reference | Identifiers | Additional information |
|---|---|---|---|---|
| Strain (*Mus musculus*) | STING ki (STING N153S knock-in mice) | *Luksch et al., 2019* | | C57BL/6 N background, not the same mouse line as *Warner et al., 2017* |
| Strain (*Mus musculus*) | Ifnar1$^{-/-}$ | *Siedel et al., 2020* | | backcrossed 20 times to C57BL/6 |
| Strain (*Mus musculus*) | Casp1$^{-/-}$ | *Reinke et al., 2020* | | C57BL/6 background |
| Antibody | Anti-Iba1, rabbit polyclonal | Fujifilm Wako Chemicals | Cat# 019–19741 | 1:1000 |
| Antibody | Anti-GFAP, chicken polyclonal | Abcam | Cat# ab4674 | 1:2000 |
| Antibody | Anti- NFκB p65 (D14E12), rabbit polyclonal | Cell Signaling Technology | Cat# 8242 | 1:500 |
| Antibody | Anti- Tubulin βIII, rabbit polyclonal | Covance | Cat# PRB-435P | 1:5000 |
| Antibody | Anti-phospho-a-synuclein, rabbit monoclonal | Abcam | Cat# ab51253 | 1:1000 |
| Antibody | Anti-a-synuclein, mouse monoclonal | BD Bioscience | Cat# 610787 | 1:2000 |
| Antibody | Anti- TH, sheep, polyclonal | Pel-Freeze | Cat# P40101 | 1:2000 |
| Antibody | Anti-synapsin, chicken monoclonal | Synaptic Systems | Cat# 160002 | 1:1000 |
| Antibody | Anti-homer, rabbit monoclonal | Synaptic Systems | Cat# 106006 | 1:1000 |
| Antibody | Anti-ASC, rabbit monoclonal | Adipogen | Cat# AG-25B-0006-C100 | 1:700 |
| Antibody | Anti-IL1b, rabbit monoclonal | Cell Signaling Technology | Cat# 2022 | 1:300 |
| Antibody | Anti- NLRP3, mouse monoclonal | Adipogen | Cat# AG-20B-00060006 | 1:1000 |
| Antibody | Anti- NeuN, chicken polyclonal | Synaptic Systems | Cat# 266006 | 1:1500 |
| Antibody | Anti- HDAC1, mouse, monoclonal | Thermo Fisher | Cat# MA5-1807 | 1:1000 |
| Antibody | Anti- IRF3, rabbit polyclonal | Santa Cruz | Cat# sc-9082 | 1:1000 |
| Antibody | Anti- GAPDH, mouse, monoclonal | Millipore | Cat# MAB374 | 1:5000 |
| Antibody | HRP conjugated donkey anti-mouse polyclonal | Jackson ImmunoResearch | Cat# 715-035-150 | 1:5000 |
| Antibody | HRP conjugated donkey anti-rabbit polyclonal | Jackson ImmunoResearch | Cat# 711-035-152 | 1:5000 |
| Antibody | Alexa 555 conjugated donkey anti-rabbit polyclonal | Invitrogen | Cat# A31572 | 1:2000 |
| Antibody | Alexa 647 conjugated donkey anti-chicken polyclonal | Jackson ImmunoResearch | Cat# 703-605-155 | 1:2000 |
| Antibody | Alexa 488 conjugated donkey anti-sheep polyclonal | Invitrogen | Cat# A11015 | 1:2000 |
| Antibody | Alexa 488 conjugated donkey anti-rabbit polyclonal | Invitrogen | Cat# A21206 | 1:2000 |
| Sequence-based reagent | Ifi44 fw primer | This paper | PCR primer | *AACTGACTGCTC GCAATAATGT* |
| Sequence-based reagent | Ifi44 rev primer | This paper | PCR primer | *GTAACACAGCA ATGCCTCTTGT* |
| Sequence-based reagent | Mx1 fw primer | This paper | PCR primer | AACCCTGCTACCTTTCAA |
| Sequence-based reagent | Mx1 rev primer | This paper | PCR primer | AAGCATCGTTT TCTCTATTTC |

*Appendix 1 Continued on next page*

*Appendix 1 Continued*

| Reagent type (species) or resource | Designation | Source or reference | Identifiers | Additional information |
|---|---|---|---|---|
| Sequence-based reagent | Sting1 fw primer | This paper | PCR primer | CTGCTGACATAT ACCTCAGTTG |
| Sequence-based reagent | Sting1 rev primer | This paper | PCR primer | GAGCATGTTGT TATGTAGCTG |
| Sequence-based reagent | Cxcl10 fw primer | This paper | PCR primer | CCAAGTGCTGC CGTCATTTTC |
| Sequence-based reagent | Cxcl10 rev primer | This paper | PCR primer | GGCTCGCAGG GATGATTTCAA |
| Sequence-based reagent | Tnfa fw primer | This paper | PCR primer | CCTGTAGCCC ACGTCGTAG |
| Sequence-based reagent | Tnfa rev primer | This paper | PCR primer | GGGAGTAGACA AGGTACAACCC |
| Sequence-based reagent | Casp1 fw primer | This paper | PCR primer | GCTGCCTGCCC AGAGCACAAG |
| Sequence-based reagent | Casp1 rev primer | This paper | PCR primer | CTCTTCAGAGTCTCTTACTG |
| Sequence-based reagent | Il1b fw primer | This paper | PCR primer | GAAATGCCACC TTTTGACAGTG |
| Sequence-based reagent | Il1b rev primer | This paper | PCR primer | TGGATGCTCTCA TCAGGACAG |
| Sequence-based reagent | Hprt1 fw primer | This paper | PCR primer | TCAGTCAACGGG GGACATAAA |
| Sequence-based reagent | Hprt1 rev primer | This paper | PCR primer | GGGGCTGTACT GCTTAACCAG |
| Sequence-based reagent | Rpl13 fw primer | This paper | PCR primer | AGCCTACCAGAA AGTTTGCTTAC |
| Sequence-based reagent | Rpl13 rev primer | This paper | PCR primer | GCTTCTTCTTCC GATAGTGCATC |
| Sequence-based reagent | Eef2 fw primer | This paper | PCR primer | CCGACTCCC TTGTGTGCAA |
| Sequence-based reagent | Eef2 rev primer | This paper | PCR primer | AGTTCAGGTCG TTCTCAGAGAG |
| Sequence-based reagent | Ym1 fw primer | This paper | PCR primer | CATTCAGTCAG TTATCAGATTCC |
| Sequence-based reagent | Ym1 rev primer | This paper | PCR primer | AGTGAGTAGCAGCCTTGG |
| Sequence-based reagent | Il4 fw primer | This paper | PCR primer | AGATGGATGTGCC AAACGTCCTCA |
| Sequence-based reagent | Il4 rev primer | This paper | PCR primer | AATATGCGAAGCA CCTTGGAAGCC |
| Sequence-based reagent | Nos2 fw primer | This paper | PCR primer | CTGCTGGTGGTGAC AAGCACATTT |
| Sequence-based reagent | Nos2 rev primer | This paper | PCR primer | ATGTCATGAGCAAA GGCGCAGAAC |
| Sequence-based reagent | Retnl1 fw primer | This paper | PCR primer | TGGAGAATAAGGTCAAGGAAC |
| Sequence-based reagent | Retnlb rev primer | This paper | PCR primer | GTCAACGAGTAAGCACAGG |
| Commercial assay or kit | Pierce Micro BCA Protein-Assay-Kit | Thermo Fisher Scientific Inc | Cat# 23235 | |
| Commercial assay or kit | WesternBright Chemilumineszenz Substrat Sirius | Biozym Scientific GmbH | Cat# 541019 | |
| Commercial assay or kit | M-MLV Reverse Transriptase Kit | Promega | Cat# M1701 | |

*Appendix 1 Continued on next page*

*Appendix 1 Continued*

| Reagent type (species) or resource | Designation | Source or reference | Identifiers | Additional information |
|---|---|---|---|---|
| Commercial assay or kit | SV Total RNA Isolation System | Promega | Cat# Z3101 | |
| Commercial assay or kit | GoTaq qPCR Master Mix | Promega | Cat# A6002 | |
| Chemical compound, drug | PhosSTOP phosphatase inhibitor | Sigma-Aldrich | Cat# 4906845001 | |
| Chemical compound, drug | Precision Plus Protein WesternC Protein Standards | BioRad | Cat# 1610376 | |
| Chemical compound, drug | RNasin Ribonuclease Inhibitor | Promega | Cat# N2511 | |
| Chemical compound, drug | Deoxynucleotide Triphosphates (dNTPs) | Promega | Cat# U1205 | |
| Chemical compound, drug | M-MLV 5 x Reaction Buffer | Promega | Cat# M1701 | |
| Chemical compound, drug | Fluoromount-G | Southern Biotech | Cat# 0100–01 | |
| Chemical compound, drug | NucSpot Live 650 Nuclear Stain | Biotium | Cat# 40082 | |
| Chemical compound, drug | Thioflavin S | Sigma-Aldrich | Cat# T1892 | |
| Chemical compound, drug | Mitosox | Thermo Fisher | Cat# M36008 | |
| Chemical compound, drug | Di(Acetoxymethyl Ester) (6-Carboxy-2',7'-Dichlorodihydrofluorescein Diacetate) (DCFH-DA) | Thermo Fisher | Cat# C2938 | |
| Software, algorithm | GraphPad Prism 5. 01 and 9.0.0 | GraphPad Software | | |
| Software, algorithm | ImageJ Fiji | Wayne Rasband (NIH) | | |
| Software, algorithm | QuantStudio5 qPCR Data Analysis Software | Thermo Fisher Scientific Inc | | |
| Other | QuantStudio 5 Real-Time PCR System | Thermo Fisher Scientific Inc | | |
| Other | Zeiss Axio Observer Z1 | Carl Zeiss | | |
| Other | Zeiss Spinning Disc | Carl Zeiss | | |
| Other | Infinite 200 M Multimode-Plate-Reader | Tecan Group | | |

