## [Editor Report]

STING has recently been implicated in the pathogenesis of neurodegenerative diseases. In the current work, the authors analysed a mouse model that expresses the constitutively active STING variant N153S and demonstrated that activation of the STING pathway via downstream signalling by type I interferons (Ifnar1-/-) or inflammasome activation (Casp1-/-) is sufficient to induce degeneration of dopaminergic neurons. Moreover, STING activation is sufficient to induce α-synuclein pathology in Parkinson's disease. This finding is of interest to the broad readership of *eLife*.

---

## [Decision Letter]

[Editors' note: this paper was reviewed by Review Commons.]

Thank you for submitting your article "Constitutively active STING causes neuroinflammation and degeneration of dopaminergic neurons in mice" for consideration by *eLife*. Your article has been reviewed by 3 peer reviewers at Review Commons, and the evaluation at *eLife* has been overseen by a Reviewing Editor and Mone Zaidi as the Senior Editor.

The revised manuscript has addressed most of the raised concerns in the previous review. Though additional data substantially strengthen the manuscript, two of the referees still remain concerned with some of the data, and they both raised some further concerns.

For instance, review #2 noted some issues with the new data in Figure 6F-I. "The authors present ASC staining to quantify ASC puncta within microglia and astroglia from striatal sections. Given Iba1+ cells have more total ASC is the increase ni ASC puncta just dut to higher overall levels i.e. if normalised for expression of ASC are there more puncta? Also an explanation for why Casp1 KO have less ASC and ASC puncta is required. Caspase1 is downstream of ASC puncta formation so why is there less ASC and ASC puncta in these KO cells?". Reviewer #2 felt that their hypothesis is largely confirmed in this revision, and support publication. But before that they should perform MAP2 staining and add some description regarding unexpected findings in Discussion. But s/he complained that "However, some points, which are within my main claims in previous submission, are not yet completely resolved. In Figure 5, pSTAT-positive cells are neurons (this is consistent with their assumption), and NF-κB positive cells also seem non-astrocyte and non-microglia. It is a little bit strange that they did not use the neuronal marker MAP2 for this analysis. They should do this, because brain is a complex tissue, and it is usually unacceptable not to dissect the pathology to cell types in a standard- or high-level journal."

If you feel that you could address these concerns, we would expect your resubmission in the near future!

---

## [Author Response]

Reviewer #1:1) “The authors could consider qualifying the observations as preliminary as no mechanistic data or longer-term pathophysiology is investigated. Indeed, the latter is well beyond the current scope and may require generation of cell-type specific STING ki mice.”

Thank you for the comment. We have qualified our observations as preliminary (line 662). Indeed, generating cell-type specific STING ki mice is part of our future plans.

2) The authors consistently write "NF-κB/inflammasomes" – these two pathways (although related) are quite distinct and should not be lumped together in such a way.

Thank you for this important note, we now corrected the text (for example see section headings in the Results section, lines 338 and 415).

3) Line 79: "NRLP3" should be corrected to NLRP3. Line 210: age of "adult mice" in weeks should be state in the text and figure legend.

Thank you, corrected.

4) Line 262: In Figure 3B and D the images look very different and there is no indication of what a positive inclusion is? This should be indicated on the image.

Thank you for the suggestion. We replaced the corresponding panels with new images, where we show the nuclei with blue, and the Thioflavin S staining with magenta pseudo-color (current Figure 3E). We marked the outline of Thioflavin S positive cells with yellow. An inset showing the magnification of some neurons with inclusions is also presented.

5) Line 280: The data of Ifi44 should also be mentioned in the text.

Thank you. We performed new experiments to show the gene expression changes in the striatum and in the substantia nigra, therefore majority of the gene expression data from the cortex has been moved to the supplementary material (supplemental Figures 3 and 5), and is not discussed in detailed in the current manuscript.

6) Line 290: Figure 4, Examining IL-1B and Caspase-1 transcripts is not a readout of inflammasome activation. pro-IL1B is upregulated in response to NFkB activity. Inflammasome activation is commonly examined in other methods e.g. via ASC puncta formation (imaging based), active IL-1B secretion (ELISA), Caspase-1 and IL-1B cleavage via western blot

Thank you for this suggestion, we performed new experiments and added the data as Figure 6.

1) We performed Western blot analysis to detect IL-1β cleavage and NLRP3 proteins from the striatum (Figure 6C-E). 2) We quantified the number of ASC puncta within microglia and astroglia from striatal sections (Figure 6F-I). 3). 3) We also measured the protein levels of several additional immune mediators in the striatum of STING ki and KO animals (supplemental Figure 6, summary heatmap is on Figure 7A).

7) Line 310: The NF-κB subunit examined should be stated (p65?). Furthermore, IRF3 translocation might be a better readout for STING activation.

We indeed detected the p65 subunit of NF-κB (antibody is listed in the supplemental Table), and now it is also indicated in the text (line 366).

We also performed subcellular fractionation and quantified IRF3 in the nuclear and cytoplasmic fractions. The data is now added on Figure 6A, B.

8) Discussion: Given the findings here suggest a strong role for NF-κB, a short discussion of IFN vs non-IFN responses from STING should be included. There have been a number of seminal papers demonstrating the importance of non-IFN STING responses of late as well as much evidence from SAVI mice to suggest some non-IFN driven pathologies.

Thank you for the suggestion. The data on inflammasomes were given a separate section in the results (from line 395). In the discussion, from line 535 we discuss the IFN dependent response and from line 548 we discuss the non-IFN driven pathways.

9) Discussion: Is there any evidence from the human SAVI patients of neuroinflammation etc. This should be mentioned either way in the discussion.

Thank you for this comment. The manifestation of neurological symptoms is not a core feature of the human SAVI disease. Some patients suffer from various neurological symptoms e.g. calcification of basal ganglia, spastic diplegia and episodes of seizure (Fremond et al., 2021).

We inserted a short text in discussion (lines 532-534).

10) Discussion: There is a large body of work demonstrating STING-induced cell death in numerous cell types. Despite this it is not mentioned nor discussed but should be. It could represent how dopaminergic neurons are lost in the STING ki mice.

Thank you for pointing out the gap in our discussion. We added additional text in lines 604-618.

11) The resolution/quality of some of the imaging is not great but this may be due to PDF compressionReviewer #2:1) The authors base their conclusions (line 215-216) on the neuroinflammatory status of their mice strongly on an assessment of the Iba1 and GFAP-positive area fraction. Increase of Iba1 and GFAP areas does not necessarily correlate with an increased cytokine production and release by the cells. Therefore, in addition the measurement of cytokine mRNAs it would be necessary to measure cytokines also on protein level (see also #4 and #5).

Thank you for this suggestion, we measured the protein levels of several immune mediators with LEGENDplex assay from the striatum, and the new data are included as Figure 7A and supplemental Figure 6.

2) In the same context: Is the increase of Iba1 and GFAP- covered area due to increased proliferation of microglia and astrocytes or due to increased expression of these markers in activated glia? How is the number of Iba1/GFAP-positive cells affected?

We quantified the number of glia cells in the striatum and in the substantia nigra of adult STING WT and STING ki mice, and, parallel with higher immunoreactivity for the corresponding markers, we detected increased number of cells as well. The quantifications are now included in supplemental Figure 1.

3) Nowadays we know that microglia and astrocytes can exist in a variety of activated states which can be either beneficial or detrimental. An analysis of disease-associated microglial markers (Keren-Shaul et al. 2017) would give a good picture of the state microglia are in.

Thank you for the suggestion. In addition to the panel of immune modulators at the protein level (supplemental Figure 6), we performed qPCR analysis of additional “M1” marker (Nos2) and additional “M2” markers (Il4, Fizz2, Ym1) (Gong et al., 2019). The data is included in Figure 7A and shown in supplemental figure 6. The findings are described from line 431.

4) It also would be of interest to determine which cell type is responsible for the observed neurodegeneration. Which cytokines are released by microglia or astrocytes upon STING activation? Even in vitro experiments would help here to get a more profound understanding.

We agree with the suggestion, however, the further in vitro experiments are beyond the scopes of this study and will be the basis of a future project.

5) In line 273 the authors describe that STING is known to activate NFkB and the inflammasome. As proof that this is also occurring in their mouse, they perform qPCR analysis of whole brain IL-1b, TNF-a and Casp1 expression. While this analysis indicates that there is indeed an increased mRNA production of proinflammatory cytokines in the brains of STING ki mice, it does not give any indication whether the inflammasome is active or not. The inflammasome is a protein complex largely regulated on protein level. Meaning an assessment of the cleavage of Caspase 1 on protein level or the presence of cleaved IL-1b in comparison to uncleaved Pro-IL-1b by Western Blot as well as a staining for the number of inflammasomes would be required to draw these conclusions.

Thank you for the suggestion. We performed additional experiments: (1) Western blot to detect pro-IL1b and IL1b and NLRP3 proteins from the striatum (Figure 6C-E), and (2) we quantified the number of ASC puncta within microglia and astroglia from striatal sections (Figure 6F-I).

6) To conclude that NFkb/inflammasome pathway is the most active/crucial in astrocytes (line 354) a staining for ASC inflammasomes would be of importance, especially as astrocytes normally do not express NLRP3.

Thank you for this comment. We stained brain sections for ASC specks and for microglia (Iba1) and astroglia (GFAP) markers (Figure 6F-I). Although amount of ASC specks in astroglia was lower than in microglia, we found still a substantial amount of ASC specks in astroglia in the brains of STING ki animals.

7) As already shown for ALS (Yu et al., 2020) and Parkin KO (Sliter et al. 2018), the authors want to further assess the relevance of the STING pathway to PD (line 27-28). Therefore, an in-depth analysis of key PD hallmarks beyond phosphorylated a-synuclein, lossthe other was parkin/PINK related (so TDP deleted) of TH-stained neurons and dopamine reduction is needed. In the discussion the authors hypothesize that autophagy (line 467) may be linked to the observed phenotype. Therefore, assessment of autophagy/mitophagy as well as mitochondrial dysfunction and mtDNA should be analysed. In the same line of thought it would be important to know if and how the observed dopamine reduction effects mouse behaviour, thus mice should be subjected to the Rotarod or pole or beam walk test.

Thank you for these suggestions. In the work by Yu et al. and Sliter et al., the STING pathway was shown to mediate neurodegeneration resulting from TDP-43 pathology and mitochondrial damage. Our work is complementary by investigating the effects of constitutive activation of STING. We have therefore focused on the signaling pathways downstream of STING. As mentioned above, the most important next step will be to separate the contributions of neuronal and glial cells by generating cell type specific STING activation. Of course, it will be interesting to see at a later time point whether STING activation feeds back. We also speculate that STING activation may also cause TDP-43 pathology. Yet, this will be part of a future study. To acknowledge that the pathology is not specific to α-synuclein, we added a short statement from line 634.

With respect to the comprehensive analysis of the PD phenotype, our work includes the classical parameters of TH neuron number, TH fiber density, dopamine concentration and synuclein pathology. With respect to mouse behavior, we note that the STING ki mice have severe inflammation in the lung, kidney and other (peripheral) organs, reduced body weight and reduced lifespan (Luksch et al., 2019; Motwani et al., 2019; Siedel et al., 2020). Motor deficits cannot be attributed to dopamine neuron degeneration and for this reason were not included (stated in the Discussion, lines 624-625). In order to expand the description of the PD phenotype we now included measurements of cytosolic reactive oxygen species, mitochondrial oxygen species and nitric oxide, which result from inflammation and are known to affect dopaminergic neurons (new Figure 8).

Reviewer #3:1) The method for quantification of TH-positive cells is not sufficient. They just described how they stained every fifth sections but did not mention how they count. This is a critical point and they should carefully provide information more than just referring their previous paper.

Counting of dopaminergic neurons and quantification of fibers was described in a dedicated section of the methods. This section has now been expanded (from line 154).

2) It is not persuasive that they did not investigate local inflammation in SN. They presented increased microglia and astrocytes in the striatum but not analyzed these cells in SN.

Indeed, we measured neuroinflammation in the substantia nigra as well, however, although increased in STING ki mice, it was less pronounced than neuroinflammation in the striatum. We now include the quantification of area fraction as well as cell number counting of microglia and astroglia in the substantia nigra of STING WT and STING ki animals (supplemental Figure 1), and also the expression of inflammatory mediators in Figure 4.

3) In Figure 3, they analyzed α-synuclein phosphorylation and β-sheet structure in the striatum. This is funny from the aspect of Parkinson's disease, which dominantly affects SN. They should perform similar experiments with SN samples. In a different aspect, the aggregates detected by Thio S may not be α-synuclein and could be tau, TDP43 or other substances. Phospho-synuclein of course does not mean aggregation, so they can consider electron microscopy.

We agree with the reviewer. To complement our data, we therefore performed solubility assay both from the striatum and from the substantia nigra to quantify the ratio of α-synuclein in the Triton X-100 soluble and insoluble fractions (Figure 3C, D) as previously (Szego et al., 2022; Szegő et al., 2019). Additionally, we quantified phosphorylated α-synuclein from the substantia *nigra a*s well (Figure 3A,B).

We also agree with the reviewer that the presence of Thioflavin S-positive inclusions may also contain other, β-sheet forming proteins and noted this from line 634.

4) Figure 5, pSTAT3 increased in Iba1-negative cells, which seem neurons from the size of nuclei. First, the authors should investigate the identity of pSTAT3-positive cells with GFAP and MAP2. If pSTAT3 is actually increased in neurons, what does it mean in the pathology? For instance, in viral infection, STAT3 activation triggers suicide of neurons to prevent further proliferation of viral particles in neurons. Is it homologous or other function?

Thank you for this suggestion. The brain sections were stained for Iba1 and GFAP. pSTAT3 nuclear staining indeed increased in non-glia cells, based on the morphology, we think in neurons. However, detailed characterization of the signal is out of the scopes of this (preliminary) study.

5) In Figure 6 and overall, cell types in which the activation of three signaling pathways, were mixed up and hard to understand the actual situation in the brain.

In our model, STING is activated in all cells. Consequently, we cannot determine the origin of immune mediators found elevated in the STING ki mice. This will require cell type specific STING activation. In order to react to the reviewer’s comment and be clearer, we have added more details about the brain region and age of mice used for each analysis also in the figures.

6) In the method section, the original paper for generation of heterozygous STING N153S KI mice should be Warner et al., JEM 2017.

We used a STING N153S ki mouse strain that was independently generated in the Technical University Dresden (Luksch et al., 2019).

7) NF-κB stains seem located in cytoplasm in Figure 5B.

We agree: especially in the young STING ki mice, cytoplasmic NF-κB staining is increased compared to STING WT mice. To quantify nuclear translocation, however, we counted the number of those cells where NF-κB signal was overlapping with the nuclear Hoechst staining.

8) In Figure 4 and 6, why the authors evaluate gene expressions in frontal cortex instead of SN or striatum.

As noted in several comments, we show here that the STING-induced pathology involves dopaminergic neurons, but believe that it is not specific for the dopaminergic system given that STING-ki is ubiquitously expressed. For practical reasons, we have used cortical samples for the expression analysis. For consistency, we now performed additional qPCR measurement from the striatum and from the substantia nigra and included them as new Figure 4 and supplemental Figure 6N-Q. The previous data from the cortex was moved to the supplemental Figures 3 and 5. Additionally, we measured the levels of several inflammatory modulators from the striatum of STING ki and KO animals (Figure 7A and supplemental figure 6A-M).

9) In some groups (Sting-ki;ifnar1-/- in Figure 6C, 6E), the values were separated to two groups, which makes readers to doubt on soundness of their genotyping.

Our genotyping protocol is highly standardized, and the genotype of the animals were correctly assigned. In Author response image 1 we provide an example of gel images showing the products after PCR reactions for the STING N153S allele (Author response image 1), STING WT allele (Author response image 1), Ifnara WT allele (Author response image 1) and lack of Ifnara allele (Author response image 1) of the same animals. We note that a bimodal distribution of phenotypes is often observed in *Ifnar-/-* mice.

**Author response image 1. sa2fig1:** Example for correct genotyping of animals used in the experiments. (a) PCR product by using primers specifically detecting the N153S variant of STING. (b) PCR product by using primers specifically detecting the WT (mouse) variant of STING. Mice with ID 66, 69, 70 and 71 are STING ki expressing both the mutant Sting1 and the WT Sting1. Mice with ID 67, 68, 72, 73 are STING WT. (c) PCR product by using primers specifically detecting the mouse Ifnar. (d) PCR product by using primers specifically detecting the lack of mouse Ifnar. All mice are Ifnar^-/-^.

[Editors' note: further revisions were suggested prior to acceptance, as described below.]

The revised manuscript has addressed most of the raised concerns in the previous review. Though additional data substantially strengthen the manuscript, two of the referees still remain concerned with some of the data, and they both raised some further concerns.For instance, review #2 noted some issues with the new data in Figure 6F-I. "The authors present ASC staining to quantify ASC puncta within microglia and astroglia from striatal sections. Given Iba1+ cells have more total ASC is the increase ni ASC puncta just dut to higher overall levels i.e. if normalised for expression of ASC are there more puncta?

In response to this comment, we measured the mean fluorescence intensity in the ASC channel from the same images that we used for ASC puncta analysis. Indeed, the intensity of the ASC signal was higher in Iba1-positive microglia than in GFAP-positive astroglia. This difference was observed in both genotypes, but particularly pronounced in STING ki animals. The data was added as the new supplemental Figure 5J and noted in line 424ff. As hypothesized by the reviewer, the increase in ASC staining intensity correlated strongly with the increase in ASC puncta as in author response image 2.

Also an explanation for why Casp1 KO have less ASC and ASC puncta is required. Caspase1 is downstream of ASC puncta formation so why is there less ASC and ASC puncta in these KO cells?".

Inflammasomes are multiprotein complexes which are formed and activated upon recognition of pathogen-associated molecular patterns (PAMPs) or damage-associated molecular patterns (DAMPs). Formation of inflammasomes results in caspase-1 activation followed by IL-1b release. Inflammasome formation in mice requires priming via NF-κB to induce expression of the scaffold protein NLRP3 (Bauernfeind et al. J Immunol 183 (2) 2009; Broz & Dixit. Nature Rev Immunol 16 (7) 2016). This priming is reduced in *Casp1*-/- mice due to the reduction of IL-1b cleavage and release and, hence, less NF-κB activation. In conclusion, inflammasome activation is reduced in caspase-1 deficient mice.

This background was clarified in line 431ff. of the text.

Reviewer #2 felt that their hypothesis is largely confirmed in this revision, and support publication. But before that they should perform MAP2 staining and add some description regarding unexpected findings in Discussion. But s/he complained that "However, some points, which are within my main claims in previous submission, are not yet completely resolved.In Figure 5, pSTAT-positive cells are neurons (this is consistent with their assumption), and NF-κB positive cells also seem non-astrocyte and non-microglia. It is a little bit strange that they did not use the neuronal marker MAP2 for this analysis. They should do this, because brain is a complex tissue, and it is usually unacceptable not to dissect the pathology to cell types in a standard- or high-level journal."

This is correct. In response to the reviewer, we stained striatal sections for pSTAT3, NeuN as neuronal marker and Iba1 for microglia. Indeed, the majority (app. 85%) of the nuclear pSTAT3 signal colocalized with NeuN. Conversely, almost 70% of striatal neurons were positive for pSTAT3 in STING ki animals. The new data is included as supplemental Figure 4 and added to the results (line 377ff).